# How to Correctly do Semantic Backpropagation on Language-based Agentic Systems

## Abstract

Language-based agentic systems have shown great promise in recent years, transitioning from solving small-scale research problems to being deployed in challenging real-world tasks. However, optimizing these systems often requires substantial manual labor. Recent studies have demonstrated that these systems can be represented as computational graphs, enabling automatic optimization. Despite these advancements, most current efforts in Graph-based Agentic System Optimization (GASO) fail to properly assign feedback to the system's components given feedback on the system's output. To address this challenge, we formalize the concept of *semantic backpropagation* with *semantic gradients*—a generalization that aligns several key optimization techniques, including reverse-mode automatic differentiation and the more recent TextGrad by exploiting the relationship among nodes with a common successor. This serves as a method for computing directional information about how changes to each component of an agentic system might improve the system's output. To use these gradients, we propose a method called *semantic gradient descent* which enables us to solve GASO effectively. Our results on both BIG-Bench Hard and GSM8K show that our approach outperforms existing state-of-the-art methods for solving GASO problems. A detailed ablation study on the LIAR dataset demonstrates the parsimonious nature of our method.

## 1 Introduction

Language-based agentic systems are being hailed as a major breakthrough in artificial intelligence, with real-world deployment well underway and numerous companies already being founded based on this technology (e.g., Pythagora-io (2023) ). Such agentic systems typically consist of multiple components. These components are selected to perform specific tasks, such as question answering, implementing and executing computer programs, or performing web searches (Wang et al., 2024; Guo et al., 2024). Due to the strength of Large Language Models (LLMs) in doing a wide array of tasks, agentic systems typically have most of their key components rely on querying LLMs. This results in communication between the components of such systems being handled with free-form natural language (Zhuge et al., 2023). However, while relying on LLMs does partially alleviate the engineering burden of building such systems, designing agentic systems remains nontrivial.

Agentic systems are often modeled as computational graphs, with components involving frozen large models having auxiliary optimizable parameters (Zhuge et al., 2024). When the graph topology is fixed, the challenge of optimizing these parameters to enable an agentic system to solve a specific problem can be modeled as the Graph-based Agent System Optimization (GASO) problem.

Famous methods that attempt to solve the GASO problem include DSPy prompt optimization methods (e.g., COPRO) (Khattab et al., 2024), GPTSwarm's node optimization (Zhuge et al., 2024), Textgrad (Yuksekgonul et al., 2024), and Trace's OptoPrime (Cheng et al., 2024). DSPy prompt optimization methods and GPTSwarm's node optimization both do so by optimizing a prompt template for each node given the input-output pairs of each node that were observed during predictions. In contrast, both TextGrad and OptoPrime adopt a backpropagation-inspired approach, wherein both of them attempt to assign a feedback to each of the components in a system by message passing in the reverse topological order (backward) of the graph given an output feedback. However, OptoPrime fails to build a compact and explicit representation of how each component can be improved to pass backward. On the other hand, TextGrad omits neighborhood nodes while computing the backward

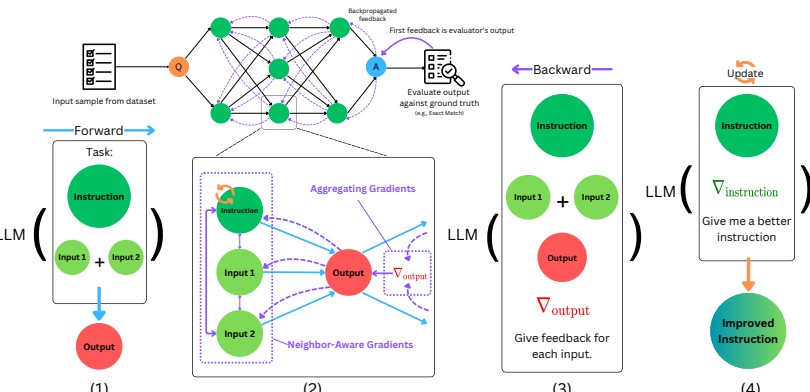

Figure 1: The entire process of our proposed LLM-based solution to GASO. Given a sample query to optimize over, (1) the forward pass of each node can be executed by joining an instruction alongside other inputs to process. Then, (2,3) the semantic gradients are generated through semantic backpropagation that crucially takes into account the neighboring nodes. And finally, (4), the semantic gradients accumulated are joined with the optimizable parameter (e.g., the instruction) and an optimization meta-prompt to retrieve update in the direction given by the semantic gradients.

messages. We argue that the aforementioned drawbacks of OptoPrime and TextGrad significantly hinder the effectiveness of backpropagation and reverse-mode automatic differentiation.

To address the issues with TextGrad and other proposed GASO solutions, we propose *semantic backpropagation* over *semantic gradients*. Semantic gradients generalize mathematical gradients by representing directional information in any semantically interoperable form, indicating how a variable in a system would change to improve the overall system performance. Semantic backpropagation serves to align TextGrad with reverse-mode automatic differentiation (Linnainmaa, 1970; 1976), which is also known as backpropagation (Werbos, 1982) in the context of neural network optimization. [1] We further propose *semantic gradient descent* which uses semantic gradients to update the optimizable parameters, and therefore solve the GASO problem. Our overall approach to solving the GASO problem is summarized in Figure 1.

We apply semantic backpropagation and semantic gradient descent to BIG-Bench Hard (BBH) (Suzgun et al., 2023) and GSM8K (Cobbe et al., 2021), finding that semantic gradient descent outperforms TextGrad, OptoPrime, and COPRO on these benchmarks. We also perform an extensive ablation study on the LIAR (Wang, 2017) dataset, showing that the method is parsimonious.

Our contributions can be summarized as follows: We **(1)** formalize the Graph-Based Agentic System Optimization (GASO) problem. **(2)** Introduce Semantic Gradients, Semantic Gradient Descent, and Semantic Backpropagation, demonstrating how these methods resolve challenges in existing GASO solvers. **(3)** Show improved performance over other GASO solvers, including COPRO, OptoPrime, and TextGrad when evaluating on BBH and GSM8K with a general question-answering setup. **(4)** Perform an ablation study on the LIAR dataset that highlights a decrease in performance when key components of our methods are removed.

## 2 BACKGROUND

### 2.1 THE GRAPH-BASED AGENTIC SYSTEM OPTIMIZATION PROBLEM

The Graph-based Agentic System Optimization (*GASO*) problem aims to optimize a system capable of delivering precise answers to a variety of user queries through a structured computational approach. We formalize this system using a directed acyclic computational graph $(V, E, H, \Theta)$. In this graph, $V$ represents the set of vertices or nodes, with each node $v \in V$ being a variable within the

---

[1]The method proposed in this paper is not to be confused with semantic backpropagation in Genetic Programming (Pawlak et al., 2014) where input-output pairs are propagated which, while different, both pass messages backward, in different semantic spaces.

system. $E$ denotes the set of directed edges between nodes, $H$ is a set of forward functions assigned to certain nodes, and $\Theta \subset V$ consists of variables that are optimizable parameters[2].

For any vertex $v$ in the graph, let $\mathrm{Predecessors}(v)$ denote its predecessor vertices and $\mathrm{Successors}(v)$ denote its successor vertices. If a node $v \in V$ has no predecessors, it either holds a specified user query $Q$ or an optimizable parameter $\theta \in \Theta$. Conversely, if a node has predecessors, it contains the result of computations performed by the function $h_v \in H$ on its predecessors, expressed as:

$$\forall v \in V \text{ s.t. } \mathrm{Predecessors}(v) \neq \emptyset, \; v = h_v(\mathrm{Predecessors}(v)). \tag{1}$$

The final response $A$ to the user query $Q$ is produced by a special output node where $\mathrm{Successors}(v) = \emptyset$. The notation $A(Q, \Theta)$ highlights the functional relationship between the graph output and the user query-parameter pair. To obtain the final response, each node of the graph is executed in topological order.

The objective in the GASO problem is to find a set of parameters $\Theta^*$ that minimizes the expected loss $\mathbb{E}_{Q \sim D}[l(Q, A(Q, \Theta^*))]$ over a distribution of queries $D$. Here, $l$ represents a loss function (e.g., negative utility) defined for a query and its corresponding response. In addition to the loss $l(Q, A)$, a semantic alphanumeric feedback $F(Q, A)$ is provided as an additional signal for optimization. In this paper, we focus on cases where variables are represented in free-form natural language; that is, $Q, A, \Theta$, and the outputs of $h_v$ are composed of alphanumeric strings. To effectively process the semantic content, the functions $h_v$ often require querying LLMs. For instance, consider a variable $v_a$ linked to its predecessors $v_q$ (a user query) and $v_\theta$ (an additional optimizable instruction that can affect the response to $v_q$). The function for $v_a$ could be expressed as $h_{v_a}(v_q, v_\theta) = \mathrm{LLM}(v_q \oplus v_\theta)$, where $\oplus$ denotes the concatenation operator, and the LLM function returns an LLM response. In practical applications, $v_\theta$ is often a prompt prefix or suffix optimized to improve the LLM's response to the task. In the general case, forward functions could take more complex forms, such as accessing a file, executing a command line and reading the result, and managing some internal thought (Wang et al., 2024; Zhang et al., 2024; Jin et al., 2024).

## 2.2 REVERSE-MODE AUTOMATIC DIFFERENTIATION

When the forward functions and the loss function are differentiable, first-order optimization methods (see the work of Beck (2017)) compute the gradient of the loss with respect to all optimizable parameters to solve GASO. Reverse-mode automatic differentiation (Linnainmaa, 1970; 1976), or RMAD, facilitates this by computing the gradient of each variable in accordance with the chain rule:

$$\frac{\partial l}{\partial v} = \sum_{w \in \mathrm{Successors}(v)} \frac{\partial l}{\partial w}\frac{\partial w}{\partial v} = \sum_{w \in \mathrm{Successors}(v)} \frac{\partial l}{\partial w}\mathbb{J}_{h_w^v}, \tag{2}$$

where $h_w^v$ represents the function $h_w$ with variables $\mathrm{Predecessors}(w) \setminus \{v\}$ fixed at their current values. Here, $\mathbb{J}$ denotes the Jacobian of $h_w^v$ with respect to $v$. The numerical gradients can be interpreted as a vector that could be added to the weights if one wants to improve the loss function. In the next section, we generalize the numerical gradients of RMAD to arbitrary strings, and the RMAD method to handle arbitrary forms of gradients.

## 3 METHODS

Building on the foundational work of Linnainmaa (1970; 1976), Pryzant et al. (2023), and Yuksek-gonul et al. (2024), we introduce the concept of *Semantic Backpropagation* over *Semantic Gradients*. In the context of the GASO problem, the semantic gradient of a loss function $l(Q, A(Q, \Theta))$ with respect to a variable $v$, denoted as $\nabla_v l_Q$, provides directional information on how altering $v$ can improve system performance for a query $Q$. Semantic backpropagation employs the final semantic gradient $\nabla_A l_Q$—typically derived from the answer feedback $F(Q, A)$—to generate semantic gradients $\nabla_v l_Q$ for all variables $v \in V$. Specifically, for a given variable $v$, the backward functions—acting as a generalization of the chain rule—are used to partially compute the semantic gradients

---

[2]While the GASO problem is general and accounts for both node and edge optimization within a graph, in this work, we are interested in credit assignment methods that distribute a semantic feedback to each of the node variables. Therefore, we focus on the special case where we assume that the edges are fixed and only optimize the parameters $\Theta$.

for Predecessors($v$) using the semantic gradient with respect to $v$. This procedure is systematically applied to all variables $v \in V$, proceeding in reverse topological order.

Using semantic gradients, we propose *Semantic Gradient Descent* to address the GASO problem. Generalizing from numerical gradient descent (Lemaréchal, 2012), this method involves the following iterative steps: **(1)** Sample a query $Q$ from a distribution $D$, **(2)** Apply semantic backpropagation to compute the semantic gradients $\nabla_v l_Q$ for all variables $v \in V$ if $l(Q, A(Q, \Theta))$ exceeds a specified threshold, **(3)** Use an optimizer $\phi$ on each parameter $\theta \in \Theta$, guided by its semantic gradients, to update the optimizable parameters. The subsequent sections detail the mechanisms of semantic backpropagation and semantic gradient descent.

## 3.1 SEMANTIC BACKPROPAGATION

Given a computational graph as described in Section 2, in our approach, we generalize the term $\frac{\partial l}{\partial w} \mathbb{J}_{h_w^v}$ in Equation (2) by introducing a set of backward functions $\{\hat{h}_w^v : w \in V, v \in \text{Predecessors}(w)\}$. Each backward function $\hat{h}_w^v$ serves as an analogue to the product of the derivative and the Jacobian in RMAD, extending it to arbitrary forward functions $h_w$ that might incorporate natural language. Specifically, for any query $Q$, node $w \in V$, and $v \in \text{Predecessors}(w)$, $\hat{h}_w^v$ maps the values of Predecessors($w$), $w$, and the gradient $\nabla_w l_Q$, to a direction $\nabla_v^w l_Q$, in the space of $v$. This direction represents how a change in $v$ would affect $w = h_w(\text{Predecessors}(w))$ in alignment with $\nabla_v^w l_Q$, while keeping the other predecessors fixed.

Instead of the summation over successors in RMAD, we introduce an aggregation function $\mathcal{A}_v$ that combines the set of directions $\{\nabla_v^w l_Q : w \in \text{Successors}(v)\}$ into a single semantic gradient $\nabla_v l_Q$ for each variable $v$. This generalizes the summation operator in RMAD, allowing for more flexible and problem-specific methods of combining gradients. Formally, we have:

$$\nabla_v l_Q = \mathcal{A}_v(\{\nabla_v^w l_Q : w \in \text{Successors}(v)\}), \text{ where } \nabla_v^w l_Q = \hat{h}_w^v(\text{Predecessors}(w), w, \nabla_w l_Q)$$

for all $w \in \text{Successors}(v)$. Algorithm 1 shows the procedure for backpropagation of semantic gradients. Our formulation thus extends the chain rule and RMAD to accommodate arbitrary functions and aggregation mechanisms.

---

**Algorithm 1** Semantic Backpropagation

**Require:** A computational graph with vertices $V$, a set of backward functions $\{\hat{h}_w^v : w \in V, v \in \text{Predecessors}(w)\}$, a set of gradient aggregation functions $\{\mathcal{A}_v : v \in V\}$, and $\nabla_A l_Q$, the gradient of $l(Q, A)$ for some query $Q$ with respect to answer $A$.
  **for** $v$ in ReverseTopologicalSort($V \setminus \{A\}$) **do**
    **for** $w$ in Successors($v$) **do**
      $\nabla_v^w l_Q \leftarrow \hat{h}_w^v(\text{Predecessors}(w), w, \nabla_w l_Q)$
    **end for**
    $\nabla_v l_Q \leftarrow \mathcal{A}_v(\{\nabla_v^w l_Q : w \in \text{Successors}(v)\})$
  **end for**
**Ensure:** $\{\nabla_v l_Q : v \in V\}$

---

> Implementation 1: Feedback on Optimizable Parameters
>
> For an **optimizable parameter** $\theta \in \Theta$, $\nabla_\theta^w l_Q = \hat{h}_w^\theta(\text{Predecessors}(w), w, \nabla_w l_Q)$, the direction of change of $\theta$ that would move $w$ towards to the direction $\nabla_w l_Q$ for $w \in \text{Successors}(\theta)$ is set to be the string:
>
> > *Input:*
> >     Predecessors($w$) $\setminus \theta$
> > *My output:*
> >     $w$
> > *Feedback received on my output:*
> >     $\nabla_w l_Q$.
>
> Here, the aggregator $\mathcal{A}_\theta$ is string concatenation.

In this work, the forward functions $h_v$ process semantic information from natural language inputs. Our implementation of the corresponding backward functions $\hat{h}_v^w$ involves querying an LLM to understand how the input value $v$ influences the output value $w$. This requires considering all inputs of $h_w$(i.e., $\text{Predecessors}(w)$) and the semantic gradient of the output $\nabla_w l_Q$. These backward functions can be customized as needed. For example, in this paper, we adopt a specific form of semantic gradient for all optimizable parameters $\theta \in \Theta$, where $\nabla_\theta l_Q$ is calculated as the aggregation over all successors of $\theta$, their semantic gradients, and the predecessors of $h_w$ excluding $\theta$ for all $w \in \text{Successors}(\theta)$. Implementation 1 provides an example of our approach for computing semantic gradients for optimizable parameters.

## 3.2   Semantic Gradient Descent

In this section, we first formalize the notion of parameter updating given semantic gradients, and then present in detail the procedure in which it is applied.

### 3.2.1   Parameter Update Function

Similar to numerical gradient descent, semantic gradient descent also requires a parameter update function, denoted as $\phi$. Given an optimizable parameter $\theta$ and a set of semantic gradients $G_\theta = \{\nabla_\theta l_{Q_i} : i \in \mathbb{N}, i \leq k\}$ of losses $l_{Q_i} = l(Q_i, A(Q_i, \Theta))$ for queries $Q_i$, the function $\phi$ updates the parameter value by moving $\theta$ according to $G_\theta$. Analogously to numerical gradient descent, the parameter $\theta$ is updated by applying the formula $\theta \leftarrow \theta - \alpha \sum_{i=1}^{k} \nabla_\theta l_{Q_i}$. One method to implement $\phi$ involves querying an LLM for an improved version of $\theta$, conditioned on $G_\theta$. We adopt this strategy and detail our implementation of the parameter update function in Implementation 2.

---

**Implementation 2: Parameter Update Function**

Given an optimizable parameter $\theta$ and a set of its semantic gradients $G_\theta$,

$$\phi(\theta, G_\theta) = \text{PostProc}(\text{LLM}(s)),$$

where s is the string

> *I'm trying to write a task-specific question answering assistant.*
> *My current prompt is:*
>   $\theta$
> *Here are some examples that it did not answer well:*
>   $l(G_\theta)$
> *Based on the above examples, write an improved prompt.*
> *Do not include the keyword "feedback" or any example-specific content in the prompt.*
> *Finish with the improved prompt wrapped by <prompt> and </prompt>,*

$l$ lists the semantic gradients with a prefix "## Example k" attached to the $k^{th}$ gradient, and PostProc is a post-processing function that extracts the improved prompt wrapped by <prompt> and </prompt> from the LLM response.

---

### 3.2.2   The Optimization Procedure

Given a parameterized graph, a loss function as described in Section 2, and a parameter update function $\phi$, semantic gradient descent solves the agentic graph optimization problem as follows. The optimizable parameters $\Theta$ are first initialized. Then, we iteratively execute the following steps. First, we repeatedly sample a query $Q$ and only compute the semantic gradients of $l_Q$ with respect to the variables if $l(Q, A(Q, \Theta))$ is above a certain threshold. Move to the next step if this threshold condition is met $b$ times for some batch size $b$. Second, we apply $\phi$ to each parameter-gradients pair $(\theta, G_\theta)$ to obtain an alternative parameter value $\theta'$ for each parameter $\theta \in \Theta$. Lastly, we apply an update gate[3] so that the parameters values are updated if the newly generated values outperform the

---

[3]Note that while TextGrad does not include such a mechanism, the official implementation of TextGrad does include it.

current values on a validation set. Formally, define a validation function

$$L_{\text{Val}}(\Theta) = \sum_{Q \in \text{Val}} l(Q, A(Q, \Theta)),$$

where elements of Val are samples from the query distribution $D$. We update $\Theta \leftarrow \{\phi(\theta, G_\theta) : \theta \in \Theta\}$ if $L_{\text{Val}}(\{\phi(\theta, G_\theta) : \theta \in \Theta\}) \leq L_{\text{Val}}(\Theta)$. See Algorithm 2 for details.

---

**Algorithm 2** Semantic Gradient Descent

---

**Require:** A computational graph with vertices $V$ and optimizable parameters $\Theta \subset V$, a distribution of queries $D$, a loss function $l$, a output feedback function $F$, a parameter update function $\phi$, a loss threshold $\tau$, a batch size $b$, and validation function $L_{\text{Val}}$.

  Initialize $\theta$ for all $\theta \in \Theta$.
  **while** terminate condition not met **do**
    $G_v \leftarrow \emptyset$
    **while** $|G_v| < b$ for any $v \in V$ **do**
      Sample query $Q \sim D$.
      **if** $l(Q, A(Q, \Theta)) > \tau$ **then**
        Given an output feedback $F(Q, A(Q, \Theta))$, compute semantic gradients $\nabla_v l_Q$ according to Algorithm 1 for all $v \in V$.
        $G_v \leftarrow G_v \cup \{\nabla_v l_Q\}$ for all $v \in V$
      **end if**
    **end while**
    **if** $L_{\text{Val}}(\{\phi(\theta, G_\theta) : \theta \in \Theta\}) < L_{\text{Val}}(\Theta)$ **then**
      $\Theta \leftarrow \{\phi(\theta, G_\theta) : \theta \in \Theta\}$
    **end if**
  **end while**
**Ensure:** $\Theta$

---

**Remark** Unlike numerical gradient descent, which keeps updating the solution in each iteration, our practical experience suggests that the update gate is essential to avoid the solution deviating to less favored regions. We argue that this gating process is necessary for consistent performance improvement against the always-update strategy implemented by many first-order optimization methods since there is a lack of theoretical justification for semantic gradient descent to improve. See empirical evidence for the significance of this gating in Section 5.2.

### 3.3 DIFFERENCE WITH TEXTGRAD

Here we present our difference with TextGrad (Yuksekgonul et al., 2024). Inspired by backpropagation, TextGrad aims to solve the GASO problem by propagating "textual gradients" in the reverse direction of the computational graph. A textual gradient of a variable is defined as a criticism of the variable presented in natural language. See Section 4.1 for more details on textual gradients. Given a query $Q$, TextGrad can be implemented as a special case of semantic backpropagation by having $A_v$ as an identity function and the backward functions $\hat{h}$ satisfying:

$$\hat{h}_w^v(\cdot, w, \nabla_w l_Q) = \hat{h}_w^u(\cdot, w, \nabla_w l_Q), \quad \text{and} \tag{3}$$

$$\hat{h}_w^v(\text{Predecessors}(w), w, \nabla_w l_Q) = \hat{h}_w^v(v, w, \nabla_w l_Q) \tag{4}$$

for all $u \in V, w \in \text{Successors}(v)$, and $u \in \text{Predecessors}(w)$.

We believe Equation (3) and Equation (4) are critical issues of TextGrad. These equations can be interpreted as follows: the function to compute $\nabla_v^w l_Q$ is independent of $v$; and $\nabla_v^w l_Q$—the direction of how $v$ changes that would lead to a change of $w$—does not depend on any other predecessors of $w$. These are not the case in reverse-mode automatic differentiation, where $\hat{h}_w^v$ implements $\frac{\partial l}{\partial w} \mathbb{J}_{h_w^v}$ depending on $v$, and $h_w^v$ is the function $h_w$ with predecessors other than $v$ fixed at their current value. Equations 3 and 4 implicitly impose independence and symmetry among the input variables of the forward functions, which is typically not true in computational graphs. This limitation is especially apparent in agentic graphs, where heterogeneity between neighboring nodes is of high

importance in order for complimentary and synergistic behavior to emerge (Zhuge et al., 2023). A prominent example is the importance of having system prompts that synergize well with users' prompts in a conversational LLM, which allows it to successfully approach virtually any objective (e.g., OpenAI, 2024). It is neither necessary, nor justified, to ignore nodes with a common successor during backpropagation.

Consider, for a concrete example, a variable $v_a$ with predecessors $v_q$ and $v_\theta$, where $v_q$ is a user query, $v_a$ is an answer to the query, and $v_\theta$ is an instruction that helps produce a good answer to the user query with relation $v_a = h_{v_a}(v_q, v_\theta)$ for some $h_{v_a}$. Following TextGrad's formulation, the gradient with respect to instruction $v_\theta$, would only explicitly depend on $v_\theta$ and $v_a$, and not the question $v_q$. It keeps any possibly useful information about the question unavailable when updating $v_\theta$. This could potentially lead to an instance learning an instruction $v_\theta$ that completely ignores the nature of the approached question, e.g., attempting to memorize the last suited answer as the exact instruction to follow. Moreover, when computing the gradient with respect to the question $v_q$, an identical function is applied as when computing the gradient with respect to $v_\theta$, which disregards the difference between the role of these variables in the system.

Semantic backpropagation solves this issue by incorporating dependency and heterogeneity between neighboring nodes into our formulation. In Section 5.2, we show empirical evidence that this issue can decrease the optimization performance.

## 4 RELATED WORKS

### 4.1 TEXTUAL GRADIENTS FOR PROMPT OPTIMIZATION

The concept of textual gradient was first introduced in the context of prompt optimization as "a local loss signal which contains information on how to improve the current prompt" (Pryzant et al., 2023). Furthermore, Pryzant et al. (2023) propose ProTeGi, an optimization method that improves prompt parameters that are used to instruct an LLM to produce an answer. Given some query-expected-answer pairs, ProTeGi computes a textual gradient for each pair and applies the textual gradients to optimize the prompt parameter. By refining the prompt through a series of textual gradient-informed edits, an improvement is observed over baseline methods including APE (Zhou et al., 2023) and AutoGPT (Significant Gravitas, 2023). GRAD-SUM (Austin & Chartock, 2024) extends ProTeGi by introducing a gradient summation procedure to prevent the prompt updates from being too specific to a query-expected-answer pair. GRAD-SUM also generalizes ProTeGi by incorporating an LLMs-as-a-judge module (Zheng et al., 2024; Fu et al., 2023; Chen et al., 2024) to relax the expected-answer requirement of ProTeGi. While textual gradient (Significant Gravitas, 2023), ProTeGi and GRAD-SUM introduce first-order optimization-like methods to text-based functions, these methods are not defined for variables not directly connected to the output variable. In addition, we observe an interesting analogy between optimizing textual prompts using textual gradients and optimizing numerical prompts using numerical gradients, as done by Schmidhuber (2015), where numerical prompts are used to query a world model (Schmidhuber, 1990) through a numerical interface.

### 4.2 OTHER BACKPROPAGATION-INSPIRED METHODS FOR GASO

Trace (Cheng et al., 2024) models the execution trace of computer programs as computational graphs (i.e., trace graphs). In Trace, the authors introduce OptoPrime as an optimization algorithm similar to TextGrad and semantic backpropagation that uses a feedback signal propagated backwards in the trace graph and then used for optimization. For each node $v$ in the trace graph, a subgraph that contains all nodes $u$ such that there is a path from $v$ through $u$ and then to the output node is assigned as feedback to $v$. Conceptually, this subgraph includes all computations influenced by $v$ that have an effect on the output. Then, an optimizer (typically based on large language models) leverages the subgraphs as feedback signals to improve the optimizable nodes. Although the subgraph of $v$ contains all relevant information on how $v$ influences the output, it does not directly indicate how improvements can be made (i.e., there is no gradient-like information). Without gradient-like information gained from a backpropagation-like process, an optimizer must itself implicitly try to estimate such information—a very non-trivial task. Another issue with the aforementioned subgraphs is that the size of these graphs scales linearly with the depth of the overall computational graph. This makes Trace incompatible with large graphs when the optimizer uses transformer-based language

models (Vaswani, 2017; Schmidhuber, 1992; Schlag et al., 2021). Unlike in Trace, semantic back-propagation explicitly backpropagates this gradient-like information and is not limited by the same linear scaling requirement. Zhou et al. (2024)—a concurrent work—introduces a backpropagation-like method for GASO which includes edge optimization. However, this approach is limited to chain-structured agentic systems, with edge optimization involving the addition, removal, or rearrangement of agents within the chain.

### 4.3 Other Methods for GASO

DSPy (Khattab et al., 2024) attempts to abstract executions of language model pipelines into *text transformation graphs*. Each node in such a graph is defined as a declarative module that takes a type of text as input and transforms it to another type of text as output. These nodes are parameterized by the instructions and sets of demonstrations that are prefixed to the input text before querying an LLM. DSPy has many implementations of optimizers for finding the best values for these parameters. Two such optimizers are BootstrapFewshot, which provides few-shot demonstrations for each step in the pipeline, and COPRO, which does coordinate-ascent optimization on each of the optimizable instruction prompts in the pipeline. These two optimizers are locally focused, i.e. each step is only implicitly aware of the entire pipeline.

GPTSwarm's node optimization method Zhuge et al. (2024) solves GASO when the edges are fixed, which is the focus of this paper. Compared to COPRO, in each iteration of GPTSwarm's node optimization method, each node's parameter is updated with respect to a local objective function specific to the node. Such local objective functions are not always available and require an accurate understanding of the function of each specific node. Optimizing such a local objective function also limits the possibility that a node could change its function through global optimization. On the other hand, GPTSwarm offers an edge optimization method that can be used as a complement of our method when approaching the GASO problem with optimizable edges.

## 5 Experimental Results

In this section, we evaluate semantic gradient descent on BIG-Bench Hard (BBH) (Suzgun et al., 2023), GSM8k (Cobbe et al., 2021) (for comparing with TextGrad, Trace's OptoPrime and DSPy's COPRO), BigCodeBench (BCB) (Zhuo et al., 2024), and LIAR (Wang, 2017) (for performing ablation and evaluating ProTeGi) datasets. Using these datasets allows for multi-domain benchmarking of our method. In all experiments, the number of forward computations required is significantly higher than the number of backward computations and optimizer calls. Therefore, for consideration of the cost-quality balance, unless otherwise specified, we use `gpt-4o-mini` (a relative cheap language model) when performing forward execution and `gpt-4-turbo` when executing the backward computation or the parameter update function. See Appendix B.1 for more discussion on the choice of language models.

### 5.1 General Question Answering

In the GSM8K, BBH, and BCB datasets, we do not provide any a priori information to the agentic system regarding the task (e.g., we do not tell the system whether "True" should be represented by a "1" or by the word "True"). The computational graph consists of seven variables with three optimizable parameters, initialized identical across tasks. The initialization is chosen to be generic. Specifically, the first two parameters are initialized to "Work out an intermediate step that helps solve the problem" and the last parameter is initialized to "Solve the problem" (Figure 2, a).

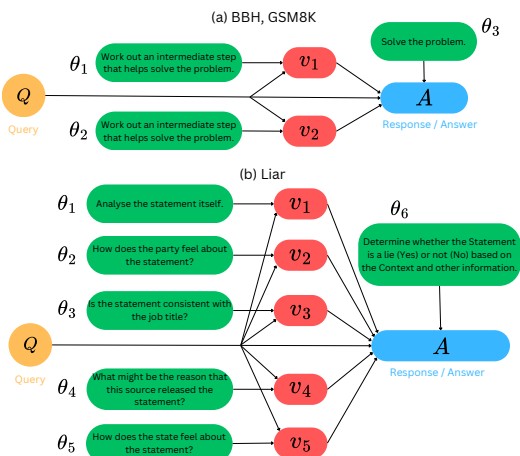

Figure 2: Initial graphs for general question answering on BBH and GSM8K (a) and LIAR (b). The variables in green (the $\theta$s) are optimizable.

**Tasks.** We experiment with the GSM8K, BBH, and BCB datasets. The GSM8K dataset includes samples of mathematical problems with a numerical answer. The BBH dataset consists of 23 tasks and 27 subtasks. Mastering all (sub)tasks requires a diverse set of skills, ranging from general mathematics to formal language processing to geometrical reasoning and more. BCB consists of diverse coding tasks, with many tasks requiring explicit function calls to external libraries.

**Baselines.** We compare against TextGrad on all three datasets, and OptoPrime on the GSM8K and BBH datasets. To make this a fair comparison, we use `gpt-4o-mini` as the LLM for the GASO loss evaluations and `gpt-4-turbo` for the backward computations and parameter updates. In all instances, we use the official implementations but with minor modifications made to allow them to work without task-specific assumptions. We detail these modifications in Appendix B.2. We also compare against the performance of OptoPrime and COPRO on the BBH dataset, as reported by Cheng et al. (2024). Due to missing implementation details, we were unable to reproduce a comparable level of performance—even when `gpt-4o-mini` was fully replaced in the implementation by `gpt-4-turbo`. We compare with the reported results on DSPy's COPRO instead of BootstrapFewshot since the latter optimizes few-shot demonstration examples, while the methods looked at here perform prompt optimization.

**Experimental Setup.** For each of the (sub)tasks, we apply our method and the baselines to a training set, and report the accuracy in a test set. For GSM8K, the training set consists of 128 randomly selected samples from the training split, and the test set is GSM8K's test split. For each of the BBH (sub)tasks, following Cheng et al. (2024), 20 samples are randomly selected as the training set, and the rest of the samples are taken as the test set. For BCB, we use the first 50 samples training split and the rest for testing. We apply four iterations of optimization with GSM8k and BBH, and 12 for BCB, where an iteration is counted when a new set of parameters is proposed. This proposal may be integrated or rejected depending on the (as introduced in Section 3.2iteration regardless. This update gate helps minimize destructive parameter updates and, in our experiments, it is computed using the training samples. In our method, we set the query distribution $D$ to be the uniform distribution on the training set. See Appendix B for more details.

**Results and Analysis.** The full experimental results are shown in Table 1. BBH results are averaged over two categories (NLP and Algorithmic), following the methodology of Suzgun et al. (2023). Semantic gradient descent performs roughly equal to TextGrad on BCB and the best in all other instances. See Appendix D for the BBH subtask-wise results. Notably, our method clearly outperforms OptoPrime as well as COPRO under BBH—even in the setting where OptoPrime and COPRO are making exclusive use of a much more expensive language model.

| Method | GSM8K | BBH NLP | BBH Algorithmic | BCB |
|---|---|---|---|---|
| **Semantic Gradient Descent** | **93.2** | **82.5** | **85.6** | **27.8** |
| TextGrad | 78.2 | 48.7 | 66.9 | 27.6 |
| OptoPrime | 83.9 | 42.3 | 50.4 | - |
| OptoPrime† | - | 75.8 | 80.6 | - |
| COPRO† | - | 73.9 | 70.0 | - |

Table 1: Average accuracy of our method on BBH (NLP and Algorithmic categories) and the GSM8K dataset. The † symbol denotes results originally reported by Cheng et al. (2024).

## 5.2 SPECIALIZED INITIALIZATION EXPERIMENTS AND ABLATION STUDY

In this section, we perform an ablation study to validate the importance of each component in the semantic gradient descent pipeline. We look at a more realistic scenario where the variables of the initial graph are highly specialized. This matches better the contemporary usage of agentic systems, where the components of the systems are assigned to implement specific and fine-grained functions (Wang et al., 2024), e.g., considering the question from a particular role's perspective (Li et al., 2023). Please also refer to Appendix A for additional ablation experiments using different network architectures and using a different forward engine.

**Task.** In the LIAR dataset (Wang, 2017), the task is to decide whether a political statement is a lie or not. Each sample in the dataset consists of five attributes, i.e., (i) the statement, (ii) the political party

of the speaker, (iii) the job title of the speaker, (iv) the state from which the speaker comes from, and (v) the source from which this statement is released. This five-attribute structure leads to an intuitive decomposition of the problem, where each component of an agentic system analyzes an attribute and then merges the analysis(Figure 2, b). This intuitive decomposition is desirable here as it allows for a relatively naive yet practically plausible agentic system architecture and prompts. Thus we can focus our evaluation on the performance of the optimizers. We use the binary classification version of LIAR as done by Pryzant et al. (2023).Here, the prefix of the response (required to be either "Yes" or "No") is used to determine how the agentic system has classified a query.

**Experiment Design.** Following the aforementioned decomposition strategy, we optimize a graph of 13 variables, of which six are optimizable parameters. These six optimizable parameters serve as instructions for an LLM. Five are initialized to guide the LLM in analyzing specific attributes of a sample, while the last parameter instructs the LLM to formulate a final answer based on the previous analyses. See Figure 2 for the visualization of the initial graph.

We compare our optimization method with four variants: (1) optimizing without semantic gradients by removing the feedback (see Implementation 1) as input of the parameter update function; (2) optimizing one parameter only (running this variant six times with a different parameter each time and reporting the average); (3) optimizing with semantic gradients computed without conditioning the neighborhood (i.e., as in Equation (3)), emulating TextGrad in our implementation; and (4) optimizing without the update gate introduced in Section 3.2, where update gate accepts parameter updates only if they performs better on a validation set.

Each variant is applied for 8 iterations. We run all the variants five times with different random seeds except for the variant that optimizes one parameter only. For this variant , we try optimizing each of the six parameters separately (once each) and report the average of these six optimizations. We optimize on 50 randomly selected samples from the LIAR training split, as done by Pryzant et al. (2023). We use these 50 random samples as both the query distribution $D$ and the validation set. Samples with missing values are filtered out since in this study we are interested in the case where each of the attributes is analyzed specifically by a component of the system.

**Results.** Table 2 shows the performance results and their respective standard errors. A noticeable drop in performance is observed when a component is removed. This supports the minimalism of semantic gradient descent. For the one instruction variant, we also compare with a best-of-$N$ method. We observe that the semantic gradient descent outperforms its one instruction variant on the best-of-$N$ metric. We report the token usage of our method in Appendix B.3.

| Method | Classification Accuracy (%) |
|---|---|
| **Semantic Gradient Descent** | **71.2 ± 3.2** |
| No Gradient | 66.0 ± 2.8 |
| One Instruction | 67.7 |
| Gradient without Neighborhood | 63.2 ± 4.1 |
| No Validation | 49.2 ± 5.0 |

Table 2: Ablation study results on the LIAR dataset. The table reports the empirical mean and standard error of the classification accuracy (i.e., the negative loss plus one). The "one instruction" variant does not have a corresponding standard error since the optimization results under this category are averaged over different optimizable parameters, which are not i.i.d. random variables.

## 6 CONCLUSION

In this work, we tackled the challenge of optimizing language-based agentic systems by introducing *semantic gradients* and *semantic backpropagation*. These concepts generalize existing credit-assignment methods, such as reverse-mode automatic differentiation and TextGrad, by incorporating neighborhood conditioning to compute directional information which can be leveraged to improve each optimizable component of the system. This framework enabled us to propose *semantic gradient descent*, effectively solving the Graph-based Agentic System Optimization (GASO) problem. Altogether, our results indicate that semantic gradients can significantly reduce the manual effort needed to optimize agentic systems, paving the way for greater scalability in AI solutions.

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

## A    ADDITIONAL ABLATIONS

### A.1    EXPERIMENTS WITH DIFFERENT ARCHITECTURES

We experiment with semantic backpropagation on GSM8k when using 2 larger graph variants, each containing 5 optimizable parameters instead of the reported 3. The first graph is organized as a network of 2x2x1 optimizable parameters, and the second is organized as a chain. The initialization is done in the same way described in Section 5.1, where every parameter is initialized to "Work out an intermediate step that helps solve the problem", except for the last parameter, which is initialized to "Solve the problem." These achieved a performance of $88.3\%$ and $88.4\%$, respectively, as compared with the original reported performance of $93.2\%$. While the reported performance is lower than that of our original architecture, it still outperforms the best results achieved by TextGrad and OptoPrime.

### A.2    EXPERIMENTS WITH DIFFERENT FORWARD ENGINES

The proposed method is expected to scale in power very closely to the underlying model, and so we would not expect it to perform well on a benchmark if backed by a weak model and using a relatively small network. Regardless, we expect to see some marginal improvements even in this case. Thus, to determine if this is indeed the case, we've run an ablation experiment with Llama3.1-8b-Instruct. We observed that our method using Llama led to a performance of $78.77\%$ on GSM8k and $55.3\%$ on BBH, compared with a performance of $77.41\%$ on GSM8k and $51\%$ on BBH using the model alone. This implies that the model is a critical part of the performance of these methods (which is unsurprising) but that our method still improves upon this.

# B    EXPERIMENTAL DETAILS

For all experiments with semantic gradient descent, we use a batch size of two and a loss threshold of 0.5. Four optimization iterations are applied for all tasks and methods we experiment in Section 5.1. See Appendix C.1 for the prompt templates we used in our implementation.

## B.1    USE OF LANGUAGE MODELS

The exact versions of language models used are `gpt-4o-mini-2024-07-18` when performing forward execution and `gpt-4-turbo-2024-04-09` when executing the backward computation or the parameter update function. `gpt-4o-mini-2024-07-18` is a language model that is more than fifty times cheaper than `gpt-4-turbo-2024-04-09` or any other version of `gpt-4-turbo`. We choose a cheaper language model for forward computation since for all competing methods in Section 5.1, the forward computation are executed tens or hundreds more times than backward computation or parameter update function. However, results reported by (Cheng et al., 2024) that are presented in Section 5.1 are based on `gpt-4-0125-preview` an early version of `gpt-4-turbo-2024-04-09` which is in the `gpt-4-turbo` family. Using an expensive language model for forward computation results in a large increase in the cost for a single experiment.

## B.2    BASELINE METHODS

**TextGrad.**    We use TextGrad's official implementation of prompt optimization with their default hyperparameters. However, TextGrad frequently fails to learn the format of tasks when a generic initial prompt such as "Answer the question" is used, resulting in a zero score in most of the (sub)tasks. To overcome this issue, we set its initial prompt to "Answer the Question. Think step by step. Finish with an answer to the question wrapped by `<answer></answer>`." with a post-processing step that extracts the content between the answer tags before evaluation.

**OptoPrime.**    We use the official implementation of Trace and its IPython Notebook example on BBH with reference to Cheng et al. (2024)'s paper. Whenever the official implementation diverges from the paper, we modify the code to rectify this divergence. There are two initializations presented by Cheng et al. (2024). We report only results using the COT initialization as this achieves stronger performance. We adopt the same update gating as in semantic gradient descent, which is also done by TextGrad's implementation. We also experimented with a version without an update gate. The removal of the update gate led to a sharp decline in performance with a score of zero on more than half the tasks.

## B.3    COMPARISON OF TOKEN COUNTS

To determine how much the inclusion of neighbouring information affects the cost of the method, we calculated the total number of tokens generated for the forward and backward passes when running the aforementioned experiment on the LIAR dataset. The results of this analysis are shown in Table 3. While the inclusion of neighbouring information has a relatively inconsequential effect on token usage, the exclusion of neighbouring information actually leads to lower token usage in the forward pass.

|                        | Neighbor | No Neighbor |
|------------------------|----------|-------------|
| Forward Input Tokens   | 273,300  | 289,850     |
| Forward Output Tokens  | 150,841  | 164,652     |
| Backward Input Tokens  | 4,182    | 3,811       |
| Backward Output Tokens | 2,006    | 1,277       |

Table 3: Comparison of token counts with and without neighbors.

# C  PROMPTS

## C.1  PROMPT TEMPLATES USED IN OUR IMPLEMENTATION

We use the string "The answer should be {desire}." for external feedback $F(Q, A)$ on the graph output $A$ given a query $Q$, where "{desire}" is a placeholder for the target output given by the dataset. Figures 3 to 6 and 8 to 10 show the Python implementation of the prompt templates used in our experiments. Specifically, Figure 3 shows the prompt templates for the forward functions for general question answering, Figure 4 shows the backward functions for general question answering, Figure 5 shows the forward functions for LIAR, Figure 6 shows the backward functions for LIAR and Figure 7 for the no neighborhood variant, Figure 8 shows the gradients with respect to optimizable parameters, Figure 9 shows a replacement of the gradients with respect to optimizable parameters for the ablation study that the gradients are not used for optimization, and Figure 10 shows the optimizer. We show the resilience of the proposed method to sensible variations of the prompts in Appendix C.4.

```
1  f"""Question:
2  {utils.add_indent(self.question.question_str)}
3  """
4              if len(self.pred_statements) != 0:
5                  prompt += \
6  f"""
7  Consider the following hints:
8  {utils.add_indent(utils.listing([statement.statement_str for statement in
       self.pred_statements]))}
9  """
10             prompt += \
11 f"""
12 Task:
13 {utils.add_indent(self.instruction.instruciton_str)}
14
15 Show your reasoning steps.
16 Finish with an output statement wrapped by <output statement> and </
       output statement>.
17 """
```

Figure 3: Prompt template of forward function for general question answering

```
1  f"""A task is performed given a question and some hints.
2
3  Task:
4  {utils.add_indent(self.instruction.instruciton_str)}
5
6  Question:
7  {utils.add_indent(self.question.question_str)}
8
9  Hints:
10 {utils.add_indent(utils.listing([statement.statement_str for statement in
      self.pred_statements]))}
11
12 Output attempt in response to the task:
13 {utils.add_indent(self.statement_str)}
14
15 Feedback on the output:
16 {utils.add_indent(utils.listing(feedback))}
17
18 Based on the feedback, how each hint should to be changed?
19 Respond one line per hint. Start with "Hint x" for the xth line.
20 """
```

Figure 4: Prompt template of backward function for general question answering

```
1   def template_context(task, problem):
2        return   """# Task
3  {task_content}
4
5  # Output format
6  Answer in no more than two sentences.
7
8  # Context
9  {text}
10
11 #Answer""".format(task_content=task, text=problem['text'])
12
13     def template_final(self, task, problem, inputs):
14        return """# Task
15 {task_content}
16
17 # Output format
18 Answer Yes or No as labels
19
20 # Context
21 {text}
22
23 # Hints
24 {input}
25
26 # Answer""".format(task_content=task, text=problem['text'], input=inputs)
```

Figure 5: Prompt template of forward function for LIAR. Function template_context is for the first five intermediate variables, while function template_final is for the answer variable.

```python
def fdbk_prompt(final_task, context, hints, answer, desire):
    return f"""A task is performed given a context and some hints
Task:
{final_task}

Context:
{context}

Hints:
{hints}

Answered: {answer}

However, the desired answer is {desire}.

How each hint needs to be changed to get the desired output? Respond one
    line per hint. Start with "Hint x" for the xth line.
"""
```

Figure 6: Prompt template of backward function for LIAR

```python
def fdbk_prompt_no_sibling(hint, answer, desire, idx):
    return f"""A task is performed given a context and some hints.
One of the hints is:
{hint}

Answered: {answer}

However, the desired answer is {desire}.

How the hint needs to be changed to get the desired output? Respond one
    line."""
```

Figure 7: Prompt template of backward function for LIAR with no neighbor information

```python
def example_str(i, input, output, fdbk):
    return f"""Input:
{utils.add_indent(input)}

My output:
{utils.add_indent(output)}

Feedback received on my output:
{utils.add_indent(fdbk)}
"""
```

Figure 8: Prompt template of the gradient with respect to optimizable parameters

```python
def example_str_no_grad(i, input, output):
    return f"""## Example {i}
Input:
{utils.add_indent(input)}

My output:
{utils.add_indent(output)}
"""
```

Figure 9: A replacement of prompt template of the gradient with respect to optimizable parameters
for ablation study when the gradient is not used for optimization.

```python
def meta_prefix(task, examples, bad_examples):
    prompt = f"""I'm trying to write a task-specific question answering
    assistant.

My current prompt is:
"{task}"
"""

    prompt += f"""\nHere are some examples that it did not answer well:
{utils.add_indent(examples)}
"""
    return prompt

def opt_prompt(task, examples, include_grad, n, is_final, bad_examples=
    True, short_prompt=False):
    prompt = meta_prefix(task, examples, bad_examples)

    prompt += f"""\nBased on the above examples, write an improved prompt
    .
Show your reasoning steps.
Do not include the keyword "feedback" or any example-specific content in
    the prompt.
Finish with the improved prompt wrapped by <prompt> and </prompt>{
    short_prompt}.
"""
    return prompt
```

Figure 10: Prompt template for the optimizer. opt_prompt returns the optimization prompt.

## C.2 EXAMPLES OF PROMPT EVOLUTION

An example showing how the optimization results diverge from an identical initialization is presented in Figure 11. The same set of initial prompts is broadcast to all 23 subtasks of BIG-Bench Hard. An example of prompt evolution for a specific task is presented in Figure 12 using the results of the experiments in Section 5.1.

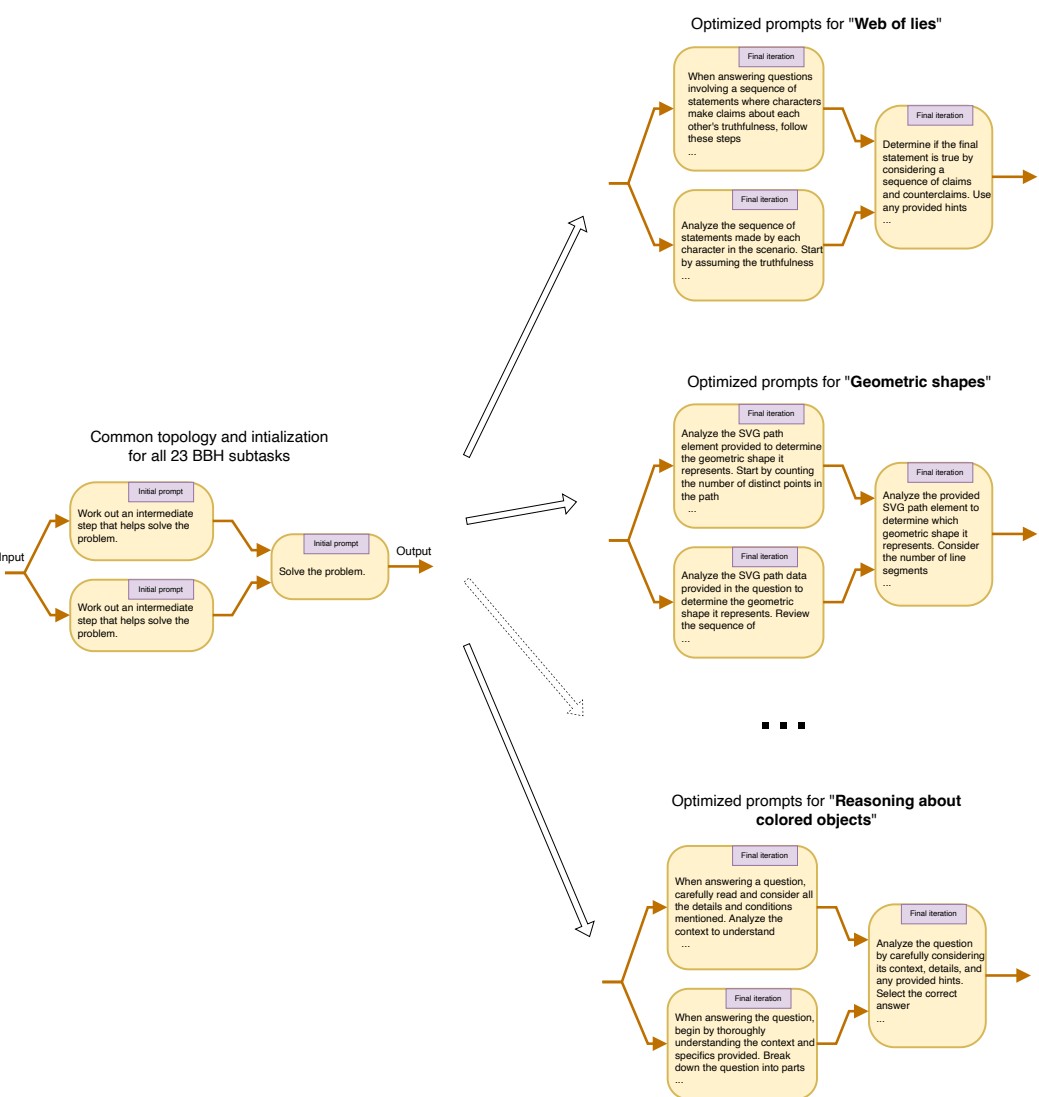

Figure 11: Processes of prompt evaluation for different BBH subtasks. The same initial prompts are set for all 23 subtasks of BIG-Bench Hard. However, the optimized prompts are specialized for each subtask.

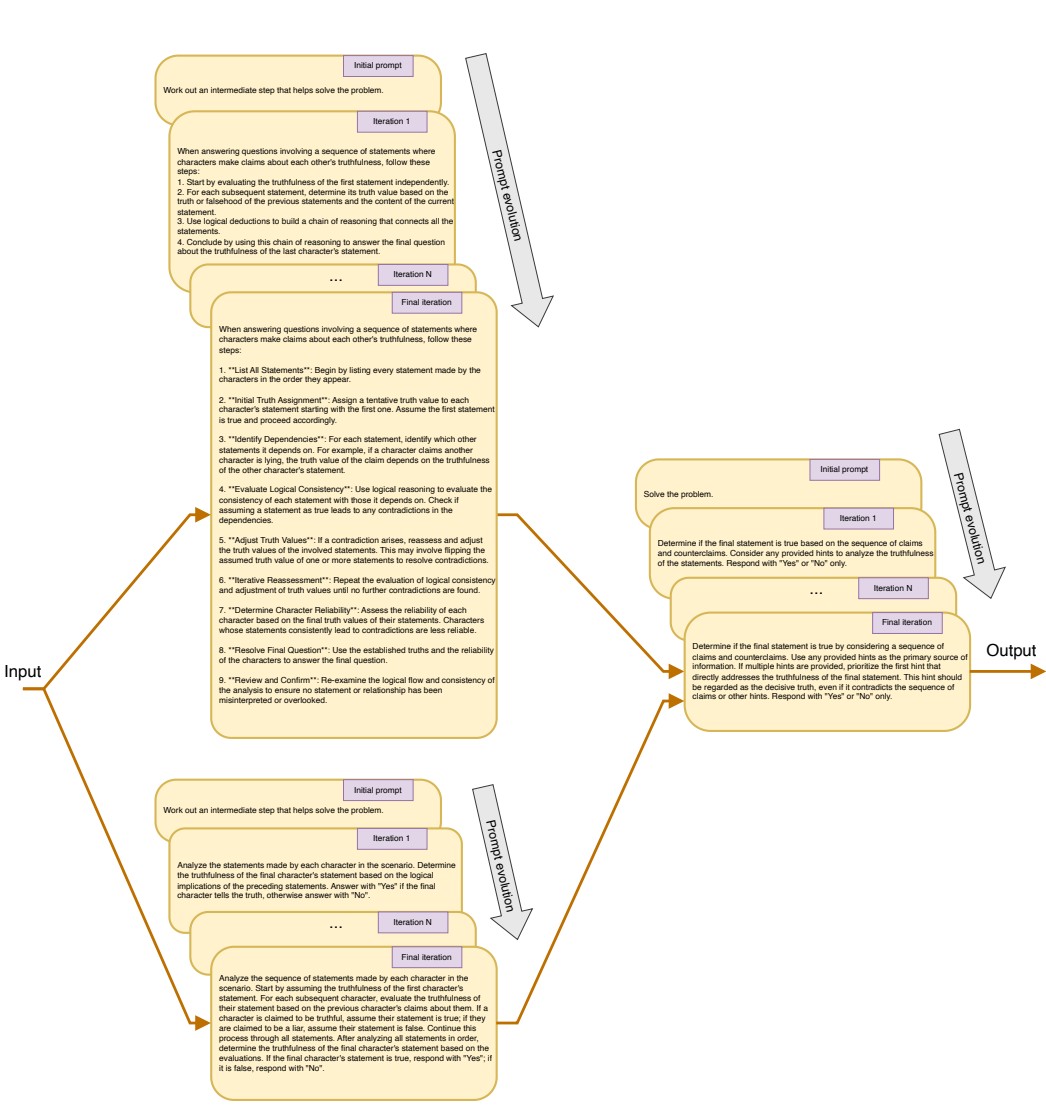

Figure 12: Evolution of prompts over the optimization iterations for "web of lies" subtask of BBH. Notice, that the initial prompts are generic, and are the same across all 23 subtasks of BIG-Bench Hard, whereas starting from iteration 1 onwards the prompts become more specialized to the target task.

### C.3 EXAMPLE OF SEMANTIC GRADIENT

We manually look at the gradients generated in the LIAR task and highlight an example that neatly demonstrates the importance of integrating neighbor information in the backward function. We can see that the information that is injected in the prompt that generates all gradients at once makes it so that important information that was generated by neighbors is taken into account when generating the feedback for the other neighbors.

In this particular example, we can see that a gradient of a neighbor that affirmatively stated a factual inaccuracy in the statement was used when generating the feedback for other neighbors. This contrasts with the no-sibling setting, where there was no way for the gradient to know that there was a factual inaccuracy.

---

FEEDBACK PROMPT WITH NEIGHBOR INFORMATION (TEMPLATE IN FIGURE 6)

A task is performed given a context and some hints

Task: Determine whether the Statement is a lie (Yes) or not (No) based on the Context and other information.

Context:

Statement: On (the federal minimum wage of) $7.75, you cant even make half the poverty level.

Job title: U.S. Representative

State: Washington state

Party: democrat

Source: an interview on MSNBC's "Politics Nation"

Hints:

1. The statement highlights that the federal minimum wage of $7.75 is insufficient to lift individuals above half the poverty level, emphasizing the inadequacy of current wage standards. This reflects a broader concern about economic inequality and the need for wage reform, particularly from a Democratic perspective.

**2. The Democratic party likely feels that the statement highlights the inadequacy of the federal minimum wage in addressing poverty, emphasizing the need for an increase to better support low-income workers. They would generally advocate for raising the minimum wage to ensure a living wage for all.**

3. Yes, the statement is consistent with the job title of a U.S. Representative, as it reflects a concern for economic issues affecting constituents. As a Democrat, advocating for higher wages aligns with party values focused on social justice and economic equity.

4. The U.S. Representative likely released the statement to highlight the inadequacy of the federal minimum wage in addressing poverty and to advocate for an increase in wages. This aligns with the Democratic Party's focus on economic justice and support for worker s' rights.

5. The state likely feels that the federal minimum wage of $7.75 is insufficient, as it does not provide a living wage and fails to meet the basic needs of individuals and families. This sentiment aligns with the Democratic Party's advocacy for raising the minimum wage to combat poverty and support workers.

Answered: No

However, the desired answer is Yes.

How does each hint need to be changed to get the desired output? Respond one line per hint. Start with "Hint x" for the xth line.

---

GENERATED GRADIENTS WITH NEIGHBOR INFORMATION

Hint 1: Clarify that the federal minimum wage is $7.25, not 7.75$, and explicitly state that the claim "you can't make half the poverty level" is factually incorrect based on federal poverty guidelines.

**Hint 2: Emphasize that while Democrats advocate for raising the minimum wage, the specific claim in the statement is demonstrably false because the federal minimum wage exceeds half the poverty level for a single individual.**

Hint 3: Indicate that the job title and party affiliation do not validate the accuracy of the statement, as the claim must be fact-checked against objective standards.

Hint 4: Reframe the intent behind the statement as an exaggeration to emphasize wage inadequacy, but note that exaggeration is not aligned with the factual context.

Hint 5: Specify that while Washington state's stance may favor higher wages, this does not negate the factual inaccuracy of the claim about making half the poverty level on the federal minimum wage.

---

FEEDBACK PROMPT WITHOUT NEIGHBOR INFORMATION (TEMPLATE IN FIGURE 7)

A task is performed given a context and some hints.

One of the hints is: **The Democratic party likely feels that the statement highlights the inadequacy of the federal minimum wage in addressing poverty, emphasizing the need for an increase to better support low-income workers. They would generally advocate for raising the minimum wage to ensure a living wage for all.**

Answered: No

However, the desired answer is Yes.

How the hint needs to be changed to get the desired output? Respond one line.

---

GENERATED GRADIENTS WITHOUT NEIGHBOR INFORMATION

**Reframe the hint to explicitly state: "The Democratic party believes the federal minimum wage must be raised to effectively combat poverty and ensure economic security for low-income workers, strongly supporting this action as a key policy priority."**

## C.4 EFFECT OF PROMPT REPHRASING

To determine the sensitivity of the proposed method to the choice of prompt, we used GPT-4o to rephrase the forward and backward prompts for the ablation experiment on the LIAR dataset reported in Table 2. The alternative forward prompts used are given in Figures 13 to 15 and the alternative backwards prompts in Figures 16 to 18.

With each of the three alternative forward prompts, the performance on the LIAR dataset was 72%, 68%, and 74%, respectively. With each of the three alternative backward prompts, the performance on the LIAR dataset was 70%, 72%, and 68%, respectively. This implies that the proposed method is relatively robust to sensible variations in the prompts.

```python
async def forward(self):
    prompt = \
f"""Context Information:
{utils.add_indent(self.question.question_str)}
"""
    if len(self.pred_statements) != 0:
        prompt += \
f"""
Hints to Consider:
{utils.add_indent(utils.listing([statement.statement_str for statement in
    self.pred_statements]))}
"""
    prompt += \
f"""
Task Description:
{utils.add_indent(self.instruction.instruciton_str)}

Explain your reasoning process in detail.
End your response with an output statement enclosed in <output statement>
    and </output statement>.
"""
    if self.is_final:
        prompt += self.question.output_format
    response = (await llm.chat(prompt))[0]
    statement = utils.parse_tagged_text(response, '<output statement>', '
    </output statement>')[0]

    self.prompt = prompt
    self.response = response
    self.statement_str = response if self.is_final else statement
```

Figure 13: First rephrased prompt template of forward function for LIAR.

```
1  async def forward(self):
2      prompt = \
3  f"""Background Context:
4  {utils.add_indent(self.question.question_str)}
5  """
6      if len(self.pred_statements) != 0:
7          prompt += \
8  f"""
9  Relevant Hints:
10  {utils.add_indent(utils.listing([statement.statement_str for statement in
        self.pred_statements]))}
11  """
12      prompt += \
13  f"""
14  Assigned Task:
15  {utils.add_indent(self.instruction.instruciton_str)}
16
17  Describe your reasoning step by step.
18  Conclude with an output statement enclosed in <output statement> and </
        output statement>.
19  """
20      if self.is_final:
21          prompt += self.question.output_format
22      response = (await llm.chat(prompt))[0]
23      statement = utils.parse_tagged_text(response, '<output statement>', '
        </output statement>')[0]
24
25      self.prompt = prompt
26      self.response = response
27      self.statement_str = response if self.is_final else statement
```

Figure 14: Second rephrased prompt template of forward function for LIAR.

```
1  async def forward(self):
2      prompt = \
3  f"""Initial Context:
4  {utils.add_indent(self.question.question_str)}
5  """
6      if len(self.pred_statements) != 0:
7          prompt += \
8  f"""
9  Hints for Consideration:
10 {utils.add_indent(utils.listing([statement.statement_str for statement in
       self.pred_statements]))}
11 """
12     prompt += \
13 f"""
14 Task Details:
15 {utils.add_indent(self.instruction.instruciton_str)}
16
17 Provide detailed reasoning for your response.
18 End with an output statement wrapped in <output statement> and </output
       statement>.
19 """
20     if self.is_final:
21         prompt += self.question.output_format
22     response = (await llm.chat(prompt))[0]
23     statement = utils.parse_tagged_text(response, '<output statement>', '
       </output statement>')[0]
24
25     self.prompt = prompt
26     self.response = response
27     self.statement_str = response if self.is_final else statement
```

Figure 15: Third rephrased prompt template of forward function for LIAR.

```
1  prompt = \
2  f"""A task is provided along with a question and several hints.
3
4  Task Description:
5  {utils.add_indent(self.instruction.instruciton_str)}
6
7  Provided Question:
8  {utils.add_indent(self.question.question_str)}
9
10 List of Hints:
11 {utils.add_indent(utils.listing([statement.statement_str for statement in
       self.pred_statements]))}
12
13 Attempted Output:
14 {utils.add_indent(self.statement_str)}
15
16 Review of the Attempt:
17 {utils.add_indent(utils.listing(feedback))}
18
19 Suggest how each hint could be improved based on the feedback.
20 Write one suggestion per hint, starting each line with "Hint x".
21 """
```

Figure 16: First rephrased prompt template of backward function for LIAR.

```
1  prompt = \
2  f"""Below is a task that requires responding to a question using provided
       hints.
3
4  Task Details:
5  {utils.add_indent(self.instruction.instruciton_str)}
6
7  Question to Answer:
8  {utils.add_indent(self.question.question_str)}
9
10 Hints Provided:
11 {utils.add_indent(utils.listing([statement.statement_str for statement in
       self.pred_statements]))}
12
13 Attempted Solution:
14 {utils.add_indent(self.statement_str)}
15
16 Feedback on the Solution:
17 {utils.add_indent(utils.listing(feedback))}
18
19 For each hint, indicate changes needed based on the feedback.
20 Respond with "Hint x" followed by the suggested improvement.
21 """
```

Figure 17: Second rephrased prompt template of backward function for LIAR.

```
1  prompt = \
2  f"""Given a task, a question, and a set of hints, complete the analysis
       below.
3
4  Task Specification:
5  {utils.add_indent(self.instruction.instruciton_str)}
6
7  Question Presented:
8  {utils.add_indent(self.question.question_str)}
9
10 Available Hints:
11 {utils.add_indent(utils.listing([statement.statement_str for statement in
       self.pred_statements]))}
12
13 Solution Attempt:
14 {utils.add_indent(self.statement_str)}
15
16 Feedback Review:
17 {utils.add_indent(utils.listing(feedback))}
18
19 Suggest improvements for each hint, referring to them as "Hint x" in your
       suggestions.
20 Provide one suggestion per hint.
21 """
```

Figure 18: Third rephrased prompt template of backward function for LIAR.

# D    TASK-WISE AND SUBTASK-WISE PERFORMANCE RESULTS

Figure 19 presents the performance results of our method, TextGrad, and OptoPrime on BBH sub-tasks and GSM8k. GPT-4o-mini is used during the forward pass of all methods and GPT-4-Turbo is used during the backward pass.

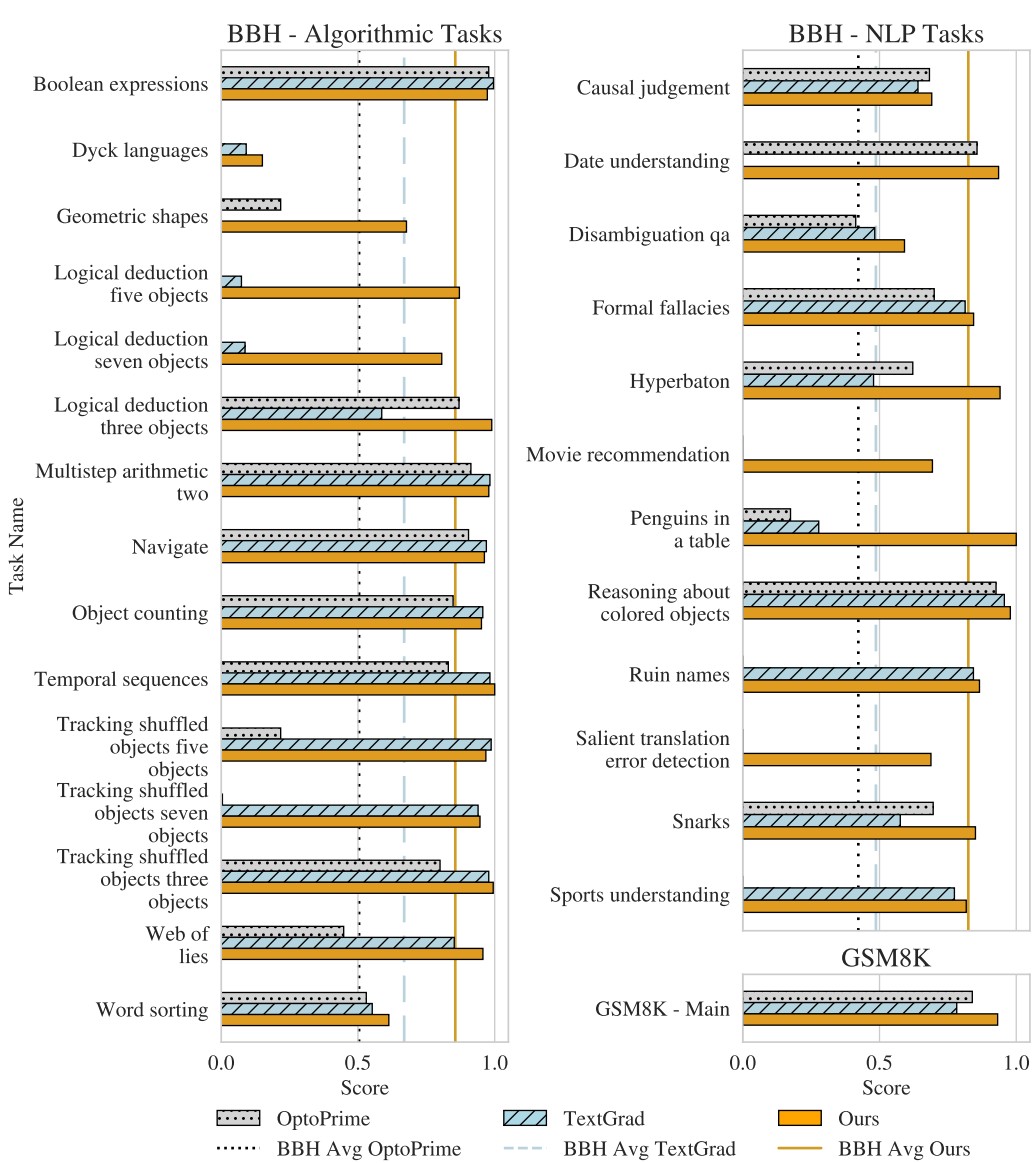

Figure 19: Scores of OptoPrime, TextGrad and semantic gradient descent (ours) on BBH and GSM8K. Semantic gradient descent outperforms both OptoPrime and TextGrad on the majority of BBH subtasks as well as GSM8K. In BBH subtasks "Movie recommendation" and "Salient transla-tion error detection" semantic gradient descent is the only one that achieves a non-zero score.

