# OpenReview forum: "How to Correctly Do Semantic Backpropagation on Language-based Agentic Systems"
_ICLR.cc/2025/Conference — Submitted to ICLR 2025_

### Official Review · Reviewer_WCzZ · 2024-10-31

**Soundness:** 3
**Presentation:** 2
**Contribution:** 2
**Rating:** 6
**Confidence:** 3

**Summary:**

Recently, Language-based agentic systems have shown great potential in solving real-world tasks. However, these works usually relied on human-designed settings (e.g., instruction or prompt). How to enable an automatic optimization for agentic systems has been a challenge. In this paper, authors introduce a semantic backpropagation with semantic gradients, which explores how to change each component of an agentic system to improve outputs. Experimental results on BIG-Bench and GSM8K demonstrate that the proposed method can solve GASO problems.

**Strengths:**

This paper formulates agentic system as a computational graph task and then introduces a semantic backpropagation to optimize each component of agentic system via semantic gradient.

**Weaknesses:**

**Weaknesses**

1. Although authors claim the difference between TextGrad as TextGrad use textual gradients while the proposed method is semantic gradients, it seems the proposed method still heavily relied on language features and human design (e.g., Implementation 2).
2. Authors claim that in TextGrad ignores the dependency and heterogeneity, can you provide some real cases about the nodes with dependencies and why they are failed in TextGrad? Does it matter in optimizing agentic system? Can you highlight more insights about your paper when compared with TextGrad?
3. The proposed method still relied on an evaluation system to obtain feedback. So what happen if applying this method to some open scenarios. Besides, this paper only evaluates Big-Bench and GSM8k. Can you try more datasets to validate the generalization of the proposed method like GAIA. MMLU?

**Questions:**

1. Can you provide some examples about semantic gradients when compared with TextGrad. From the provided code, the semantic gradient seems is still the natural language form.

---

> ### Author Response · Authors · 2024-11-25
> **Response to WCzZ (1/3)**
>
> We thank the reviewer for their time and detailed feedback. We have addressed each of your comments below and modified the paper accordingly.
>
> ---
>
> > Although authors claim the difference between TextGrad as TextGrad use textual gradients while the proposed method is semantic gradients, it seems the proposed method still heavily relied on language features and human design (e.g., Implementation 2).
>
> Thank you for raising this point. While semantic gradients are a general framework that can accommodate both numerical and textual gradients, our experiments focus on tasks requiring natural language, which allows for direct comparison with methods like TextGrad. You are right to point out that we still need some human input for designing the prompts in forward and backward functions. However, our method is robust to this choice of prompts, as demonstrated in the following experiment.
>
> We have rerun our experiment on the LIAR dataset with three different rephrasings of the forward and backward prompt templates, which were generated using GPT-4o. The mean and standard deviation of the scores of these variations are reported below (and included in the paper):
>
> | Variation           | Accuracy (%)    |
> |---------------------|-----------------|
> | Baseline            | 71.2 $\pm$ 3.2  |
> | Backward            | 71.3 $\pm$ 2.5  |
> | Forward             | 70.0 $\pm$ 1.6  |
>
>
> As you can see, while there is some impact on the performance by varying the template (as would be expected of any hyperparameter), our method remains robust to changes here.
>
>
>
> > The proposed method still relied on an evaluation system to obtain feedback. So what happen if applying this method to some open scenarios.
>
> We agree with you that the method still relies on an evaluation system. This framework is used to provide an accurate and stable evaluation of a family of methods designed for solving the GASO problem. Applying these methods to open-ended tasks is an exciting direction of work; however, it is orthogonal to the scope of this paper, which aims to introduce a general and sound method for the GASO problem. There are several ways in which semantic backpropagation could be applied to open-ended tasks without an explicit evaluation system, such as using language models directly to provide feedback [1]. We leave this investigation for future work.
>
> [1] Zheng 2023. Judging LLM-as-a-Judge with MT-Bench and Chatbot Arena.
>
>
> > Besides, this paper only evaluates Big-Bench and GSM8k. Can you try more datasets to validate the generalization of the proposed method like GAIA. MMLU?
>
> Thank you for bringing attention to this important point. We conducted an additional experiment using BigCodeBench. Here, the base model (GPT-4o-mini) achieves a 25.3% solve rate. In comparison, our model with a 2x1 variable architecture achieves a 27.8% solve rate, indicating that it has been able to perform optimization. When using a simpler architecture of only a single variable, we observe a performance of 28.07%. This is similar to the performance of TextGrad (27.6%). It is important noting that the single variable optimization case with no neighbor variables is the special case where semantic backpropagation and Textgrad are identical. Altogether, this indicates that BigCodeBench is likely most well suited to either a single variable system or a much larger number of variables outside the scope of the networks we have looked at in this work. We have added this result to the paper.

---

> ### Author Response · Authors · 2024-11-25
> **Response to WCzZ (2/3)**
>
> > Authors claim that in TextGrad ignores the dependency and heterogeneity, can you provide some real cases about the nodes with dependencies and why they are failed in TextGrad? Does it matter in optimizing agentic system? Can you highlight more insights about your paper when compared with TextGrad? Can you provide some examples about semantic gradients when compared with TextGrad. From the provided code, the semantic gradient seems is still the natural language form.
>
> We address this comment by manually looking at the gradients generated in the LIAR task and highlight an example that neatly demonstrates the importance of integrating neighbor information in the backward function. We can see that the information that is injected in the prompt that generates all gradients at once makes it so that important information that was generated by neighbors is taken into account when generating the feedback for the other neighbors.
>
> In this particular example, we can see that a gradient of a neighbor that affirmatively stated a factual inaccuracy in the statement was used when generating the feedback for other neighbors. This contrasts with the no-sibling setting, where there was no way for the gradient to know that there was a factual inaccuracy.
>
> This is an excellent suggestion that we think significantly improved the clarity of our presentation, and we thank the reviewer for bringing attention to it.
>
> For the sake of clarity, we first describe the templates used (which can be found in Figures 6 and 7 in the paper) in more detail:
>
> For an optimizable parameter $\theta \in \Theta$, $\nabla_\theta^w l_Q = \hat h_w^\theta(Predecessors(w), w, \nabla_w l_Q$), the direction of change of $\theta$ that would move $w$ towards to the direction $\nabla_w l_Q$ for $w\in Successors(\theta)$ is set to be the string:
>
> """
>
> Task: {query}
>
> Context:
> {$\mathcal{A}(Predecessors(w) \setminus \theta$)}
>
> Answered: {$w$}
>
> However, {$\nabla_w l_Q$}
>
> How does each hint need to be changed to get the desired output?
> Respond one line per hint. Start with "Hint x" for the xth line.
>
> """
>
> Here, the aggregation function $\mathcal{A}$ concatenates all the inputs by enumerating them as "Hints", and $\nabla_w l_Q$ is a simple string that describes what the desired target answer is (either "the desired answer is Yes" or "the desired answer is No").
>
> In the case where the neighborhood is not taken into account, only one predecessor of $w$ is given in the prompt, which will follow the following template:
>
> """
>
> A task is performed given a context and some hints.
>
> One of the hints is: {$v$}
>
> Answered: {$w$}
>
> However, {$\nabla_w l_Q$}
>
> How the hint needs to be changed to get the desired output? Respond with one line.
>
> """
>
> where $v \in Predecessors(w)$.
>
> ---
>
> The following comment contains the example extracted from the LIAR experiment (and added to the paper in Section C.3). The architecture used for this experiment is described fully in Figure 2.

---

> ### Author Response · Authors · 2024-11-25
> **Response to WCzZ (3/3)**
>
> -------- Neighborhood Backward Prompt --------
>
> A task is performed given a context and some hints
>
> Task:
> Determine whether the Statement is a lie (Yes) or not (No) based on the Context and other information.
>
> Context:
>
> Statement: On (the federal minimum wage of) \$7.75, you cant even make half the poverty level.
> Job title: U.S. Representative
> State: Washington state
> Party: democrat
> Source: an interview on MSNBC's "Politics Nation"
>
> Hints:
> 1. The statement highlights that the federal minimum wage of \$7.75 is insufficient to lift individuals above half the poverty level, emphasizing the inadequacy of current wage standards. This reflects a broader concern about economic inequality and the need for wage reform, particularly from a Democratic perspective.
> 2. **The Democratic party likely feels that the statement highlights the inadequacy of the federal minimum wage in addressing poverty, emphasizing the need for an increase to better support low-income workers. They would generally advocate for raising the minimum wage to ensure a living wage for all.**
> 3. Yes, the statement is consistent with the job title of a U.S. Representative, as it reflects a concern for economic issues affecting constituents. As a Democrat, advocating for higher wages aligns with party values focused on social justice and economic equity.
> 4. The U.S. Representative likely released the statement to highlight the inadequacy of the federal minimum wage in addressing poverty and to advocate for an increase in wages. This aligns with the Democratic Party's focus on economic justice and support for worker s' rights.
> 5. The state likely feels that the federal minimum wage of \$7.75 is insufficient, as it does not provide a living wage and fails to meet the basic needs of individuals and families. This sentiment aligns with the Democratic Party's advocacy for raising the minimum wage to combat poverty and support workers.
>
> Answered: No
>
> However, the desired answer is Yes.
>
> How does each hint need to be changed to get the desired output?
> Respond one line per hint. Start with "Hint x" for the xth line.
>
> -------------------------------------------------------
> -------- Generated Gradients --------
>
> Hint 1: Clarify that the federal minimum wage is _\\$7.25, not \\$7.75_, and explicitly state that the claim "you can't make half the poverty level" is factually incorrect based on federal poverty guidelines.
>
> **Hint 2: Emphasize that while Democrats advocate for raising the minimum wage, the specific claim in the statement is demonstrably false because the federal minimum wage exceeds half the poverty level for a single individual.**
>
> Hint 3: Indicate that the job title and party affiliation do not validate the accuracy of the statement, as the claim must be fact-checked against objective standards.
>
> Hint 4: Reframe the intent behind the statement as an exaggeration to emphasize wage inadequacy, but note that exaggeration is not aligned with the factual context.
>
> Hint 5: Specify that while Washington state's stance may favor higher wages, this does not negate the factual inaccuracy of the claim about making half the poverty level on the federal minimum wage.
>
> ==========================================
>
> -------- No Neighborhood Backward Prompt --------
>
> A task is performed given a context and some hints.
>
> One of the hints is:
> **The Democratic party likely feels that the statement highlights the inadequacy of the federal minimum wage in addressing poverty, emphasizing the need for an increase to better support low-income workers. They would generally advocate for raising the minimum wage to ensure a living wage for all.**
>
> Answered: No
>
> However, the desired answer is Yes.
>
> How the hint needs to be changed to get the desired output? Respond one line.
>
> -------------------------------------------------------
> -------- Generated Gradient --------
>
> **Reframe the hint to explicitly state: "The Democratic party believes the federal minimum wage must be raised to effectively combat poverty and ensure economic security for low-income workers, strongly supporting this action as a key policy priority."**
>
> ---
>
> We again thank the reviewer for their time and detailed feedback. We hope we have been able to adequately address your concerns. We remain open to further discussion.

---

> > ### Comment · Reviewer_WCzZ · 2024-11-29
> >
> > Thanks for authors' response. Most of my concerns has been addressed and I am willing to increase my score to 6. However, I think the presentation of this paper about semantic gradient should be improved. Just also mentioned by Reviewer eMLC, hy82. The presentation of this paper is somewhat confusion about semantic gradient. From my perspective, it is also difficult to distinguish the difference between the proposed method and TextGrad from the paper.  Besides, from the authors' response, the generated gradients is different from TextGrad but still as the language form (like hint). Therefore, I think authors can provide more precise description about this part in the future version as the concept of ``semantic gradient`` is really confusing.

---

> > > ### Author Response · Authors · 2024-11-29
> > >
> > > > Thanks for authors' response. Most of my concerns has been addressed and I am willing to increase my score to 6.
> > >
> > > We greatly appreciate the reviewer's engagement and are particularly grateful to them for their support for acceptance.
> > >
> > > > However, I think the presentation of this paper about semantic gradient should be improved. Just also mentioned by Reviewer eMLC, hy82. The presentation of this paper is somewhat confusion about semantic gradient.
> > >
> > > We agree that the presentation of the concept of a semantic gradient was not as clear initially as it could have been. We believe the feedback we received from all the reviewers has allowed us to significantly improve the clarity of the paper. We have included more visualizations (Figure 2 in the main text) and a concrete example (Section C.3). We have also reorganized the text such that the idea is communicated more clearly.
> > > We will endeavor to continue to improve the clarity of the work and quality of the writing even more in the camera-ready version.
> > >
> > > > From my perspective, it is also difficult to distinguish the difference between the proposed method and TextGrad from the paper. Besides, from the authors' response, the generated gradients is different from TextGrad but still as the language form (like hint). Therefore, I think authors can provide more precise description about this part in the future version as the concept of semantic gradient is really confusing.
> > >
> > > The term "semantic gradient" in the context of the GASO problem refers to the generalized framework that includes both numerical gradients, textual gradients, and semantically similar concepts (see the first paragraph of Section 3 and Section 3.3 for the differences with TextGrad; we will further improve the clarity of these sections in the camera-ready version). Such a generalization is only possible because of the compatibility with automatic differentiation, which is something that TextGrad does not offer. Unlike TextGrad, semantic backpropagation takes into account the neighborhood of a node when producing its gradient (as made explicit in equations (3) and (4) in Section 3.3 of the paper).
> > >
> > > We understand that the distinction may have seemed tied to the natural language form. However, as reviewer hy82 correctly noted:
> > >
> > > > if I understand the paper correctly, the paper is saying that if you have a function c = f(a, b), the autodiff rule for da given dc should also depend upon b, not just a, c, and dc (which is what TextGrad's formulation does).
> > >
> > > Meaning that, while TextGrad does not generalize RMAD, together, semantic gradients and semantic backpropagation serve as a generalization for both RMAD (to be extendable to text) and TextGrad (to be compatible with automatic differentiation). This is a subtle but very important difference.
> > >
> > > ---
> > >
> > > We again thank the reviewer for their feedback regarding the clarity of the presentation of semantic gradients, which has been valuable and with which we've employed to significantly improve the quality of this work.
> > > We will further improve the clarity in our camera-ready version and remain at the reviewer's disposal should they have any additional questions.

---

### Official Review · Reviewer_HxPC · 2024-11-03

**Soundness:** 3
**Presentation:** 3
**Contribution:** 3
**Rating:** 6
**Confidence:** 3

**Summary:**

To enable text optimization (omitting numerical gradients) of model outputs based on feedback, the authors propose Graph-based Agentic System Optimization (GASO). This framework formalizes the concept of semantic backpropagation with semantic gradients, generalizing several key optimization techniques, including reverse-mode automatic differentiation and TextGrad, by exploiting the relationships among nodes with a common successor. GASO demonstrates effectiveness in both the BIG-Bench Hard and GSM8K datasets.

References:

[1] TextGrad: Automatic "Differentiation" via Text.

**Strengths:**

1. This work follows an important line of research focused on text optimization based on natural language gradients.

2. The experimental results demonstrate significant improvements over existing semantic optimization methods, such as TextGrad.

3. The proposed method is clear and straightforward to implement.

**Weaknesses:**

1. The evaluation datasets are somewhat limited, comprising only BIG-Bench Hard and GSM8K. In contrast, TextGrad has been evaluated on additional datasets, including code generation, drug molecule optimization, and radiotherapy treatment plan optimization. It would be beneficial to see the potential of GASO applied to a wider range of applications.

2. Since GASO incorporates additional neighboring information in its computations, the cost associated with this approach is not clearly defined. It would be valuable to understand the trade-off between computational cost and performance.

**Questions:**

See Weakness above.

1. Performances on other reasoning tasks, such as code generation.
2. The trade-off between performance and cost.

---

> ### Author Response · Authors · 2024-11-25
> **Response to HxPC**
>
> We thank the reviewer for their time and detailed feedback. We have addressed each of your comments below and modified the paper accordingly.
>
> ---
>
> > The evaluation datasets are somewhat limited, comprising only BIG-Bench Hard and GSM8K. In contrast, TextGrad has been evaluated on additional datasets, including code generation, drug molecule optimization, and radiotherapy treatment plan optimization. It would be beneficial to see the potential of GASO applied to a wider range of applications.
>
> To address this, we conducted an additional experiment using BigCodeBench. Here, the base model (GPT-4o-mini) achieves a 25.3% solve rate. In comparison, our model with a 2x1 variable architecture achieves a 27.8% solve rate, indicating that it has been able to perform optimization. When using a simpler architecture of only a single variable, we observe a performance of 28.07%. This is similar to the performance of TextGrad (27.6%). It is important noting that the single variable optimization case with no neighbor variables is the special case where semantic backpropagation and Textgrad are identical. Altogether, this indicates that BigCodeBench is likely most well suited to either a single variable system or a much larger number of variables outside the scope of the networks we have looked at in this work. We have added this result to the paper.
>
>
> > Since GASO incorporates additional neighboring information in its computations, the cost associated with this approach is not clearly defined. It would be valuable to understand the trade-off between computational cost and performance.
>
> We agree that this is a concern. To clearly understand the trade-off in compute when incorporating neighboring information, we have reran our Liar experiments and logged the number of tokens that are processed for both the backward and forward passes for both the input and the output. The following are the statistics of these logs:
>
> |                      | Neighbor | No Neighbor |
> |----------------------|----------|-------------|
> | Total Input Tokens   | 277,482  | 293,661     |
> | Total Output Tokens  | 152,847  | 165,929     |
>
> We can see that the no-neighbor variation somewhat surprisingly uses more tokens overall. To get a better understanding of the effect of the neighboring information on compute, we can separate the forward and the backward statistics, and we find that more tokens are generated for the neighbor variation when using the backward pass:
>
> |                     | Neighbor | No Neighbor |
> |---------------------|----------|-------------|
> | Total Input Tokens  | 4,182    | 3,811       |
> | Total Output Tokens | 2,006    | 1,277       |
>
> Because the backward plays only a small role in overall computation when compared to the forward pass (only ~1.5% of the total token count), the more compute needed for incorporating neighboring information would in most cases not be a significant concern.
>
> ---
>
> We again thank the reviewer for their time and detailed feedback. We hope we have been able to adequately address your concerns. We remain open to further discussion.

---

> > ### Comment · Reviewer_HxPC · 2024-11-28
> >
> > Thank you for the clarification! I believe that point 6 accurately reflects my positive evaluation of the paper.

---

### Official Review · Reviewer_hy82 · 2024-11-04

**Soundness:** 3
**Presentation:** 3
**Contribution:** 3
**Rating:** 5
**Confidence:** 3

**Summary:**

This paper falls in the vein of research that's attempting to expand the concept of "gradient-based optimization" to language-based agentic systems. Specifically, several papers have attempted to take the concept of "autodiff" and apply it to graph-based agentic systems. From my understanding, this paper is most similar to TextGrad, but identifies some improvements in TextGrad that 1. make this paper more analogous to standard autodiff,  2. improves performance by taking advantage of more information during the backwards pass.

**Strengths:**

Overall, I think this paper identifies a reasonable gap in the existing textual gradient-based approaches (e.g. TextGrad). Specifically, if I understand the paper correctly, the paper is saying that if you have a function `c = f(a, b)`, the autodiff rule for `da` given `dc` should also depend upon `b`, not just `a`, `c`, and `dc` (which is what TextGrad's formulation does). This follows naturally from autodiff's definition, where    say, the autodiff rule for `da` in `c = mm(a, b)` involves `b`.

The "intuitive" explanation given in the paper also makes sense to me.

This motivation for their results is also validated by their experimental results.

**Weaknesses:**

I personally found the presentation of the paper somewhat confusing. In particular, I think a more concrete example walking through what exactly a semantic gradient looked like would make the paper much easier to understand, particularly if it were contrasted against the gradients from TextGrad. I see the examples of the evolved prompts in the appendix (which are useful!) but I'd be interested to see how the actual gradients look like, since that's the primary way in which this paper differs from prior work.

More broadly speaking, I also find myself being somewhat skeptical of the autodiff analogy, and to what extent semantic gradients meaningfully map to autodiff. For example, for mathematical autodiff a gradient directly corresponds to how that parameter effects the output loss (assuming an epsilon step). With semantic gradients, however, there's no such correspondence like this.

I would also be interested in seeing how different graph structures lead to different performance. For example, as far as I can tell, all the graphs in this paper involve only one "layer".

I'd also like to see more benchmarks, as opposed to just GSM8k and BBH NLP/Algorithmic.

**Questions:**

See above.

---

> ### Author Response · Authors · 2024-11-25
> **Response to hy82 (1/3)**
>
> We thank the reviewer for their time and detailed feedback. We have addressed each of your comments below and modified the paper accordingly.
>
> ---
>
> > Specifically, if I understand the paper correctly, the paper is saying that if you have a function c = f(a, b), the autodiff rule for da given dc should also depend upon b, not just a, c, and dc (which is what TextGrad's formulation does).
>
> Exactly. In this way, semantic backpropagation bridges the gap between methods like TextGrad and autodiff, finding that (somewhat unsurprisingly) the corrected method performs better.
>
>
> > More broadly speaking, I also find myself being somewhat skeptical of the autodiff analogy, and to what extent semantic gradients meaningfully map to autodiff. For example, for mathematical autodiff a gradient directly corresponds to how that parameter effects the output loss (assuming an epsilon step). With semantic gradients, however, there's no such correspondence like this.
>
> Thank you for raising this point. You are right in noting that, unlike numerical gradients, it is not guaranteed that a semantic gradient, when applied to text, will guarantee a decrease in loss. This is true even with an epsilon step in some embedding space. However, this is also true and indeed inevitable for most surrogate models that might be used instead of the true environment for evaluation feedback. What semantic backpropagation provides is a sound method for incorporating such surrogate feedback in a manner compatible with how automatic differentiation works.
>
>
> > I would also be interested in seeing how different graph structures lead to different performance. For example, as far as I can tell, all the graphs in this paper involve only one "layer".
>
> To address this, we have now run an ablation to examine whether the reported results extend to different graph structures. Specifically, we looked at the performance of semantic backpropagation on GSM8k when using 2 larger graph variants, each containing 5 optimizable parameters instead of the reported 3. The first graph is organized as a network of 2x2x1 optimizable parameters, and the second is organized as a chain. The initialization is done in the same way described in Section 5.1, where every parameter is initialized to  "Work out an intermediate step that helps solve the problem", except for the last parameter, which is initialized to "Solve the problem." These achieved a performance of 88.3% and 88.4%, respectively, as compared with the original reported performance of 93.2%. We've added these results to the paper. While the reported performance is lower than that of our original architecture, it still outperforms the best results achieved by TextGrad and OptoPrime.
>
>
> > I'd also like to see more benchmarks, as opposed to just GSM8k and BBH NLP/Algorithmic.
>
> To address this, we conducted an additional experiment using BigCodeBench. Here, the base model (GPT-4o-mini) achieves a 25.3% solve rate. In comparison, our model with a 2x1 variable architecture achieves a 27.8% solve rate, indicating that it has been able to perform optimization. When using a simpler architecture of only a single variable, we observe a performance of 28.07%. This is similar to the performance of TextGrad (27.6%). It is important noting that the single variable optimization case with no neighbor variables is the special case where semantic backpropagation and Textgrad are identical. Altogether, this indicates that BigCodeBench is likely most well suited to either a single variable system or a much larger number of variables outside the scope of the networks we have looked at in this work. We have added this result to the paper.

---

> ### Author Response · Authors · 2024-11-25
> **Response to hy82 (2/3)**
>
> > I personally found the presentation of the paper somewhat confusing. In particular, I think a more concrete example walking through what exactly a semantic gradient looked like would make the paper much easier to understand, particularly if it were contrasted against the gradients from TextGrad. I see the examples of the evolved prompts in the appendix (which are useful!) but I'd be interested to see how the actual gradients look like, since that's the primary way in which this paper differs from prior work.
>
> We address this comment by manually looking at the gradients generated in the LIAR task and highlight an example that neatly demonstrates the importance of integrating neighbor information in the backward function. We can see that the information that is injected in the prompt that generates all gradients at once makes it so that important information that was generated by neighbors is taken into account when generating the feedback for the other neighbors.
>
> In this particular example, we can see that a gradient of a neighbor that affirmatively stated a factual inaccuracy in the statement was used when generating the feedback for other neighbors. This contrasts with the no-sibling setting, where there was no way for the gradient to know that there was a factual inaccuracy.
>
> This is an excellent suggestion that we think significantly improved the clarity of our presentation, and we thank the reviewer for bringing attention to it.
>
> For the sake of clarity, we first describe the templates used (which can be found in Figures 6 and 7 in the paper) in more detail:
>
> For an optimizable parameter $\theta \in \Theta$, $\nabla_\theta^w l_Q = \hat h_w^\theta(Predecessors(w), w, \nabla_w l_Q$), the direction of change of $\theta$ that would move $w$ towards to the direction $\nabla_w l_Q$ for $w\in Successors(\theta)$ is set to be the string:
>
> """
>
> Task: {query}
>
> Context:
> {$\mathcal{A}(Predecessors(w) \setminus \theta$)}
>
> Answered: {$w$}
>
> However, {$\nabla_w l_Q$}
>
> How does each hint need to be changed to get the desired output?
> Respond one line per hint. Start with "Hint x" for the xth line.
>
> """
>
> Here, the aggregation function $\mathcal{A}$ concatenates all the inputs by enumerating them as "Hints", and $\nabla_w l_Q$ is a simple string that describes what the desired target answer is (either "the desired answer is Yes" or "the desired answer is No").
>
> In the case where the neighborhood is not taken into account, only one predecessor of $w$ is given in the prompt, which will follow the following template:
>
> """
>
> A task is performed given a context and some hints.
>
> One of the hints is: {$v$}
>
> Answered: {$w$}
>
> However, {$\nabla_w l_Q$}
>
> How the hint needs to be changed to get the desired output? Respond with one line.
>
> """
>
> where $v \in Predecessors(w)$.
>
> ---
>
> The following comment contains the example extracted from the LIAR experiment (and added to the paper in Section C.3). The architecture used for this experiment is described fully in Figure 2.

---

> ### Author Response · Authors · 2024-11-25
> **Response to hy82 (3/3)**
>
> -------- Neighborhood Backward Prompt --------
>
> A task is performed given a context and some hints
>
> Task:
> Determine whether the Statement is a lie (Yes) or not (No) based on the Context and other information.
>
> Context:
>
> Statement: On (the federal minimum wage of) \$7.75, you cant even make half the poverty level.
> Job title: U.S. Representative
> State: Washington state
> Party: democrat
> Source: an interview on MSNBC's "Politics Nation"
>
> Hints:
> 1. The statement highlights that the federal minimum wage of \$7.75 is insufficient to lift individuals above half the poverty level, emphasizing the inadequacy of current wage standards. This reflects a broader concern about economic inequality and the need for wage reform, particularly from a Democratic perspective.
> 2. **The Democratic party likely feels that the statement highlights the inadequacy of the federal minimum wage in addressing poverty, emphasizing the need for an increase to better support low-income workers. They would generally advocate for raising the minimum wage to ensure a living wage for all.**
> 3. Yes, the statement is consistent with the job title of a U.S. Representative, as it reflects a concern for economic issues affecting constituents. As a Democrat, advocating for higher wages aligns with party values focused on social justice and economic equity.
> 4. The U.S. Representative likely released the statement to highlight the inadequacy of the federal minimum wage in addressing poverty and to advocate for an increase in wages. This aligns with the Democratic Party's focus on economic justice and support for worker s' rights.
> 5. The state likely feels that the federal minimum wage of \$7.75 is insufficient, as it does not provide a living wage and fails to meet the basic needs of individuals and families. This sentiment aligns with the Democratic Party's advocacy for raising the minimum wage to combat poverty and support workers.
>
> Answered: No
>
> However, the desired answer is Yes.
>
> How does each hint need to be changed to get the desired output?
> Respond one line per hint. Start with "Hint x" for the xth line.
>
> -------------------------------------------------------
> -------- Generated Gradients --------
>
> Hint 1: Clarify that the federal minimum wage is _\\$7.25, not \\$7.75_, and explicitly state that the claim "you can't make half the poverty level" is factually incorrect based on federal poverty guidelines.
>
> **Hint 2: Emphasize that while Democrats advocate for raising the minimum wage, the specific claim in the statement is demonstrably false because the federal minimum wage exceeds half the poverty level for a single individual.**
>
> Hint 3: Indicate that the job title and party affiliation do not validate the accuracy of the statement, as the claim must be fact-checked against objective standards.
>
> Hint 4: Reframe the intent behind the statement as an exaggeration to emphasize wage inadequacy, but note that exaggeration is not aligned with the factual context.
>
> Hint 5: Specify that while Washington state's stance may favor higher wages, this does not negate the factual inaccuracy of the claim about making half the poverty level on the federal minimum wage.
>
> ==========================================
>
> -------- No Neighborhood Backward Prompt --------
>
> A task is performed given a context and some hints.
>
> One of the hints is:
> **The Democratic party likely feels that the statement highlights the inadequacy of the federal minimum wage in addressing poverty, emphasizing the need for an increase to better support low-income workers. They would generally advocate for raising the minimum wage to ensure a living wage for all.**
>
> Answered: No
>
> However, the desired answer is Yes.
>
> How the hint needs to be changed to get the desired output? Respond one line.
>
> -------------------------------------------------------
> -------- Generated Gradient --------
>
> **Reframe the hint to explicitly state: "The Democratic party believes the federal minimum wage must be raised to effectively combat poverty and ensure economic security for low-income workers, strongly supporting this action as a key policy priority."**
>
>
> ---
>
> We again thank the reviewer for their time and detailed feedback. We hope we have been able to adequately address your concerns. We remain open to further discussion.

---

> ### Author Response · Authors · 2024-12-02
> **Rebuttal Consideration**
>
> Dear reviewer hy82, as the end of the discussion period is imminent, we would like to gently remind you to please consider our recent rebuttal. We have gone to great lengths to resolve all of your concerns about our work (and such modifications to the work are expected by the ICLR reviewing process). Namely, we:
>
> 1. Improved the presentation of the paper and provided a concrete example that shows how a semantic gradient looks like.
> 2. Addressed the reviewer's concern regarding the autodiff analogy.
> 3. Added ablations on different graph structures.
> 4. Added BigCodeBench to our list of benchmarks.
>
> We are grateful for the reviewer's thoughtful and constructive feedback, which we believe significantly enhanced the quality of our work. If the reviewer feels these changes have adequately addressed their concerns, we would greatly appreciate their consideration of a revised score. If there are any remaining points of concern, we would appreciate any further feedback.
>
> Thank you for your time and effort in reviewing our work.

---

### Official Review · Reviewer_eMLC · 2024-11-05

**Soundness:** 2
**Presentation:** 2
**Contribution:** 3
**Rating:** 5
**Confidence:** 3

**Summary:**

This work presents a framework to extend the text-based gradient method (ask the model to refine the prompt. This series of works often formulates this refinement as a gradient decent process of the discrete prompt) for handling the graph structure, such as using multiple prompts to form an agent for problem-solving.

**Strengths:**

This work systematically discussed the "backpropagation" of the text-based gradient method and offered a framework to aggregate the gradients, making the text-based gradient methods close to the normal gradient descent.

**Weaknesses:**

The paper writing could be improved. I think placing figures and examples from the appendix into the main document could largely improve the readability. The experiment setting could be more clearly stated in the main document. I found those in the appendix, such as the initial graph for BBH, GSM8K, and LIAR. However, the number of nodes and the diameter of the graph are small, raising concerns about the proposed method's generalization ability on more complicated graphs, such as the MCTS, tree of thoughts, or wizardLM series.

Needs more in-depth analyses, such as the robustness of the prompt templates for the forward and backward functions. It's also better to test other models than GPT4 since the GPT4 has already achieved a very high score on GSM8K and BBH. Introducing other models could also be helpful in evaluating the backward function design. If I understand correctly, the backward prompt filled with real materials (instruction, question, response, and other statements) could be very long and complicated. My concern is that the GPT4 may not follow it well.

**Questions:**

see weakness

---

> ### Author Response · Authors · 2024-11-25
> **Response to eMLC**
>
> We thank the reviewer for their time and detailed feedback. We have addressed each of your comments below and modified the paper accordingly.
>
> ---
>
> > The paper writing could be improved. I think placing figures and examples from the appendix into the main document could largely improve the readability. ... The experiment setting could be more clearly stated in the main document. I found those in the appendix, such as the initial graph for BBH, GSM8K, and LIAR.
>
> We thank the reviewer for this comment.
> We found that Figures 2 and 3 in particular give an example of the systems we are describing in a compact way---as well as the experimental setup---and so we have now moved them into the main text.
>
>
> > However, the number of nodes and the diameter of the graph are small, raising concerns about the proposed method's generalization ability on more complicated graphs, such as the MCTS, tree of thoughts, or wizardLM series.
>
> While extending this work to MCTS or the like is very much non-trivial and far outside the scope of this work, we agree that determining the resilience to graph structure is important. To address this we have now run an ablation where we look at whether the reported results extend to different graph structures. Specifically, we looked at the performance of semantic backpropagation on GSM8k when using 2 larger graph variants, each containing 5 optimizable parameters instead of the reported 3. The first graph is organized as a network of 2x2x1 optimizable parameters, and the second is organized as a chain. The initialization is done in the same way described in Section 5.1, where every parameter is initialized to  "Work out an intermediate step that helps solve the problem", except for the last parameter, which is initialized to "Solve the problem." These achieved a performance of 88.3% and 88.4%, respectively, as compared with the original reported performance of 93.2%. We've added these results into the paper. While the reported performance is lower than that of our original architecture, it still outperforms the best results achieved by TextGrad and OptoPrime.
>
>
> > Needs more in-depth analyses, such as the robustness of the prompt templates for the forward and backward functions.
>
> To address this, we have rerun our experiment on the Liar dataset with three different rephrasings of the forward and backward prompt templates, which were generated using GPT-4o. The mean and standard deviation of the scores of these variations are reported below (and included in the paper):
>
> | Variation  | Accuracy (%) |
> |----------------|---------------|
> | Baseline | 71.2 $\pm$ 3.2  |
> | Backward | 71.3 $\pm$ 2.5  |
> | Forward | 70.0 $\pm$ 1.6  |
>
>
> As you can see, while there is some impact on the performance by varying the template (as would be expected of any hyperparameter), our method remains robust to changes here.
>
> > It's also better to test other models than GPT4 since the GPT4 has already achieved a very high score on GSM8K and BBH. Introducing other models could also be helpful in evaluating the backward function design.
>
>
> The proposed method is expected to scale in power very closely to the underlying model, and so we would not expect it to perform well on a benchmark if backed by a weak model and using a relatively small network. Regardless, we expect to see some marginal improvements even in this case. Thus, to determine if this is indeed the case, we've now run an ablation experiment with Llama3.1-8b-Instruct. We observed that our method using Llama led to a performance of 78.77% on GSM8k and 55.3% on BBH, compared with a performance of 77.41% on GSM8k and 51 on BBH using the model alone. This implies that the model is a critical part of the performance of these methods (which is unsurprising) but that our method still improves upon this.
>
> > If I understand correctly, the backward prompt filled with real materials (instruction, question, response, and other statements) could be very long and complicated. My concern is that the GPT4 may not follow it well.
>
> Indeed, long-term dependencies have historically been a significant challenge in any sequence processing task. However, the context length of these models now reaches considerable lengths, and our results show that GPT-4 can adequately handle them at this scale. For larger scales, several methods already exist that accommodate effectively arbitrary context lengths (e.g., [1], [2]).
>
> [1] Munkhdalai, T., Faruqui, M., & Gopal, S. (2024). Leave no context behind: Efficient infinite context transformers with infini-attention. arXiv preprint arXiv:2404.07143.
>
> [2] Reid, M., Savinov, N., Teplyashin, D., Lepikhin, D., Lillicrap, T., Alayrac, J. B., ... & Mustafa, B. (2024). Gemini 1.5: Unlocking multimodal understanding across millions of tokens of context. arXiv preprint arXiv:2403.05530.
>
> ---
>
> We again thank the reviewer for their time and detailed feedback. We hope we have been able to adequately address your concerns. We remain open to further discussion.

---

> > ### Comment · Reviewer_eMLC · 2024-12-02
> >
> > Thanks the authors for the detailed responses. However, I would like to keep my scores.

---

> > > ### Author Response · Authors · 2024-12-02
> > >
> > > We thank the reviewer for their response. However, we have gone to great lengths to address all of the reviewer's comments (cleaning up the writing, moving figures to the main text, adding a study showing robustness to prompt templates, testing other LLM backbones, looking at different architectures, showing context length is a non-issue). If the reviewer has no additional concerns regarding the paper, we would appreciate if the reviewer would adjust their score to reflect the current state of the paper. Otherwise, we would gently ask the reviewer for some clarifications regarding the concerns the reviewer has with the current version of the paper.

---

### Author Response · Authors · 2024-11-25
**Rebuttal Revision 25 Nov 2024, 01:14 GMT**

We would like to thank all the reviewers for their time and detailed feedback. We have done our utmost to integrate everything that has been said to improve the paper. This has taken some time to do and not been trivial. Most significantly, we have done the following:

1. Cleaned up the text in the paper to resolve grammatical errors and ambiguities.
2. Added BigCodeBench to the list of benchmarks we evaluate our method on.
3. Added ablation experiments for larger graph architectures. (Section A.1)
4. Added ablation experiments for the use of different models for the forward computations. (Section A.2)
5. Added input/output token statistics to determine how much the inclusion of neighboring information affects the cost of the method. (Section B.3)
6. Added a concrete example showcasing the significance of neighborhood information when generating the gradients. (Section C.3)
7. Added prompt robustness experiments for the forward and backward templates. (Section C.4)


Each reviewer will also find a tailored response below. Where logical, we have updated the paper in response to the feedback. Where possible, we have run an experiment to address each comment. We hope that the reviewers will find the improved paper adequate, but we of course remain open to further discussion.

---

### Author Response · Authors · 2024-11-28

We would like to thank all the reviewers for their time and insightful comments. We additionally thank HxPC for their response and their recommendation to accept the paper.

As the discussion period is drawing to a close, we would like to thank reviewers eMLC, hy82, and WCzZ for their comments that we have used to improve our submission. We would also like to gently ask them if our comprehensive responses and the corresponding changes we made to the paper are sufficient to have resolved their concerns? As always, we remain open to further discussion.

---

### Meta-Review · Area_Chair_ARS9 · 2024-12-20

**Metareview:**

This paper introduces a new framework called Graph-based Agentic System Optimization (GASO) that extends the concept of gradient-based optimization to language-based agentic systems. It follows the direction of TextGrad, and proposes "semantic backpropagation" with "semantic gradients" to optimize individual components of an agentic system. The key claim is that by incorporating information from neighboring nodes during the backward pass, this method more closely mimics standard automatic differentiation and leads to improved performance. The experimental results on BIG-Bench Hard, GSM8K, and BigCodeBench demonstrate the effectiveness of GASO over baseline methods like TextGrad. A core finding is that considering the dependencies and heterogeneity of nodes within the agentic graph leads to better optimization.

Building upon similar concepts explored in TextGrad, this paper presents a systematic approach to addressing the "backpropagation" process for text-based gradient methods. The proposed framework provides a method for aggregating gradients in a graph-based agentic system, making it closer to traditional gradient descent. The experimental results demonstrate performance improvement over existing semantic optimization methods like TextGrad on the evaluated benchmarks.This paper also identifies a valid gap in existing textual gradient approaches by highlighting the importance of considering neighbor information during the backward pass, an aspect neglected by TextGrad.

The main weaknesses of the paper primarily concern the clarity of the presentation, particularly regarding the concept of "semantic gradients" and the distinction between GASO and TextGrad, which several reviewers found confusing, prompting requests for more concrete examples and visualizations. A further limitation was the initial scope of the evaluation datasets. While the addition of BigCodeBench during the rebuttal offered some value, the experimental results demonstrated only a marginal gain over the baseline (+0.2%), failing to alleviate the concern. Another key challenge identified is the conceptual analogy to traditional autodiff, as semantic gradients lack the direct and guaranteed loss reduction characteristic of numerical gradients. Finally, the computational overhead incurred by incorporating neighboring information remains a notable concern.

Overall, this paper lacks a clear illustration on the difference between GASO and TextGrad. And the focus of this paper is on agentic systems, but the benchmark (e.g., GSM8k) doesn't inherently require the capabilities typically associated with such systems, such as planning and environmental interaction. This raises the question of whether the reported performance gains truly reflect the method's effectiveness and the focus of this paper. Therefore, given the disconnect between the paper's focus on agentic systems and the non-agentic nature of the limited evaluation benchmarks, I lean towards rejection of this paper.

**Additional Comments On Reviewer Discussion:**

Although the authors responded to reviewer queries during the rebuttal and received some acknowledgment for their efforts, the core concerns about the evaluation benchmark and the unclear differentiation between GASO and TextGrad remained unresolved, leading to the rejection of this paper.

---

### Decision · Program_Chairs · 2025-01-22

Reject